# Binary peptide coacervates as an active model for biomolecular condensates

Shoupeng Cao [1,2,5], Peng Zhou [3,5], Guizhi Shen[3], Tsvetomir Ivanov[2], Xuehai Yan [3] ✉, Katharina Landfester [2] ✉ & Lucas Caire da Silva [2,4] ✉

Biomolecular condensates formed by proteins and nucleic acids are critical for cellular processes. Macromolecule-based coacervate droplets formed by liquid-liquid phase separation serve as synthetic analogues, but are limited by complex compositions and high molecular weights. Recently, short peptides have emerged as an alternative component of coacervates, but tend to form metastable microdroplets that evolve into rigid nanostructures. Here we present programmable coacervates using binary mixtures of diphenylalanine-based short peptides. We show that the presence of different short peptides stabilizes the coacervate phase and prevents the formation of rigid structures, allowing peptide coacervates to be used as stable adaptive compartments. This approach allows fine control of droplet formation and dynamic morphological changes in response to physiological triggers. As compartments, short peptide coacervates sequester hydrophobic molecules and enhance bio-orthogonal catalysis. In addition, the incorporation of coacervates into model synthetic cells enables the design of Boolean logic gates. Our findings highlight the potential of short peptide coacervates for creating adaptive biomimetic systems and provide insight into the principles of phase separation in biomolecular condensates.

Biomolecular condensates are ubiquitous in cells and participate in vital processes such as cellular metabolism, protein modification, and stress response[1,2]. Their critical biological roles have inspired the creation of synthetic analogs to elucidate the biophysical principles and molecular mechanisms underlying their formation, properties, and functions[3,4]. The development of life-like compartments based on synthetic coacervates that undergo liquid-liquid phase separation (LLPS) similar to biological condensates, is an emerging research direction in biotechnology and synthetic biology[5-7]. In contrast to membrane-bound polymer or lipid-based vesicular compartments with a dilute interior, coacervates are typically membrane-free and molecularly crowded liquid microdroplets (1-100 μm)[6]. Coacervate droplets provide a microenvironment that can sequester and concentrate various biomolecules and ions[8-10]. This enables the control and regulation of biochemical kinetics in biomimetic applications, including enzyme and RNA metabolism, ribosome biogenesis, gated membrane transport, and membrane-mediated tandem catalysis[9,11-14].

In recent years, biomolecular condensates have been closely associated with disordered proteins with low-complexity domains, where droplet formation via liquid-liquid phase separation (LLPS) is driven by weak and multiple attractive inter/intramolecular interactions[15-17]. The prevalence and importance of biomolecular condensates in cells suggest that their prebiotic analogues may have played a role in regulating intracellular complex behaviour, a prerequisite for early life[18,19]. Protein-derived peptides are biologically relevant building blocks and key components in the formation of

[1]College of Polymer Science and Engineering, State Key Laboratory of Polymer Materials Engineering, Sichuan University, Chengdu 610065, PR China. [2]Max Planck Institute for Polymer Research, 55128 Mainz, Germany. [3]State Key Laboratory of Biochemical Engineering, Key Laboratory of Biopharmaceutical Preparation and Delivery, Institute of Process Engineering, Chinese Academy of Science, Beijing 100190, PR China. [4]Department of Chemistry, McGill University, Montreal H3A 0B8, Canada. [5]These authors contributed equally: Shoupeng Cao, Peng Zhou. ✉e-mail: yanxh@ipe.ac.cn; landfester@mpip-mainz.mpg.de; lucas.cairedasilva@mcgill.ca

coacervate droplets[20–22]. Most peptide coacervates are typically prepared by charge-driven complex coacervation, which heavily involves synthetic macromolecules such as poly-L-lysine and polyarginine, and requires other oppositely charged components such as DNA and ATP[23–25]. The well-defined composition and structure of oligopeptides and proteins allow the systematic study of the formation mechanisms and properties of synthetic coacervates, which has provided insight into the formation and dissolution principles associated with intracellular LLPS[26–29]. This approach has also guided the development of coacervate components from polypeptides and short peptides with minimal sequences[20,30–32]. Recently, short peptides with simplified structures (2-10 amino acids) and compositions have shown potential as an alternative paradigm for forming droplets via self-coacervation of single solute species. This approach significantly simplifies the construction of synthetic biomolecular condensates[33–35].

Short peptide coacervates have been used to create cell-like compartments with specific properties, such as the selective partitioning of molecules and catalysts, determined by their amino-acid composition and sequence[36,37]. Particularly, peptide coacervates have been used to create catalytic compartments that enable and regulate chemical reactions in aqueous media, offering a platform for the development of microreactors and artificial organelles[36–38]. These studies underscore the potential of short peptide coacervates for engineering synthetic compartmentalized constructs with biological complexity and life-like functions[39–41]. In addition, due to their simplified composition and tailored sequences, short peptides offer a straightforward approach to uncovering the fundamental principles and molecular grammars that govern peptide phase separation through the synthesis and investigation of easily accessible peptide libraries. However, short peptides have a strong tendency to form rigid and ordered fiber-like structures due to the extensive hydrogen bonding that promotes the alignment and arrangement of peptide chains, favouring the formation of solid aggregates in a low energy state[26,28,42–44]. The molecular understanding of the phase separation and coacervation process based on short peptides remains incomplete, making it difficult to precisely control phase separation and the properties of the resulting droplets[27,28].

Current peptide coacervates are limited to specific peptide sequences and designs, which may limit their compositional complexity, functionality, and biological relevance to intracellular condensates[34,35]. Advances in this field require further exploration of the phase separation behaviour of short peptides to develop a wider range of compositions and functionalities for more demanding applications. One interesting application of peptide coacervates is their combination with complex coacervates to create multicompartment coacervate-in-coacervate systems with distinct microenvironments resembling the hierarchical organization of organelles and cytosolic compartments in natural cells[37,45]. This may not only allow the sequestration and activation of hydrophilic species such as enzymes, but also provide an alternative approach to integrating more robust hydrophobic catalysts such as transition metal catalysts, an emerging paradigm for integrating diverse catalytic pathways into synthetic systems for biotechnological purposes[46]. However, challenges in the compositional design and stability of peptide coacervates have limited the fundamental understanding of their biomimicry role as an active model of biomolecular condensates at the molecular level, as well as the exploration of their potential to resemble biological hierarchical architecture and catalytic functions.

Here, we demonstrate programmable liquid-liquid phase separation of tripeptides to engineer active coacervates as an attractive synthetic model for biomolecular condensates. By mixing binary tripeptides in specific ratios and subjecting them to a pH-induced phase separation process, conditions can be created to selectively yield stable liquid coacervates in a highly controlled manner. Molecular dynamics simulation indicated that the mixed binary system has a lower propensity for aggregation and clustering due to fewer intermolecular hydrogen bonds between the peptides, and weaker intermolecular interactions, facilitating liquid-liquid phase separation and droplet formation. The binary peptide coacervates (BPCs) exhibited reversible phase separation, compartmentalization, and dynamic morphological transformation in response to various triggers (e.g., pH change, heating, oxidation chemistry, organic solvent exchange, and urea addition). The peptide coacervates demonstrated sequestration and partitioning capabilities for a wide range of active species, acting as microreactors that enable and enhance the catalytic activity of chemical catalysts and the synthesis of compounds of increasing complexity. To further illustrate their life-like properties, binary peptide coacervates were integrated as sub-compartments within a model membrane-bound artificial cell, resulting in a multi-compartment cell-like structure with adaptive features and catalytic properties. This setup allowed the design of Boolean logic gates (OR and AND) using chemical inputs. Our results create opportunities for the design of flexible synthetic coacervate-based biomimetic complex systems of short peptides that may be useful as model systems that can expand the understanding and interrogation of biological processes, such as the principles of phase separation in biomolecular condensates.

## Results and discussion
### Formation of coacervate droplets with short peptides

Peptides with phenylalanine dipeptide (FF) motifs are widely recognized as versatile building blocks that have been extensively studied for their role in controlling phase separation due to their tuneable composition and structures[27,42,47]. FF peptides exhibit a high propensity to form kinetically confined solid hydrogels and fibrous structures[44,48–50]. They also serve as models for understanding the aggregation processes involved in neurodegenerative diseases[51]. The properties FF-based peptides are determined by the number and strength of non-covalent interactions including hydrogen bonding, hydrophobic forces, π-π stacking, and electrostatic interactions[28,52].

Recent studies have shown that FF derivatives can also form micro-sized liquid droplets[20,36]. Initial studies have shown that the self-assembly of carboxybenzyl-protected diphenylalanine (FF) leads to the formation of supramolecular nanofibrils, with metastable liquid condensates identified as an intermediate stage[42]. This droplet formation is associated with a liquid-liquid phase separation (LLPS) process and the kinetic frustration of nanofibril development. More recently, FF derivatives with bulky capping moieties have been shown to resist the formation of fiber-like structures, likely due to steric hindrance and prevention of ordered crystalline domain formation[33]. All these early observations focused on the behaviour of solutions containing a single type of dipeptide, resulting in phase transitions determined by the association of peptides of the same type. However, there is a lack of studies on the phase behaviour of mixtures of different small peptides, which are more biologically relevant than systems containing only a single type of peptide. Furthermore, the effect of peptide composition on LLPS and fiber formation remains poorly understood. We hypothesized that increasing the diversity of FF peptide components in solution would disrupt and significantly reduce the orienting intermolecular interactions that drive fiber and aggregate formation. This disruption would help to stabilize the coacervate phase, making it a suitable biomimetic compartment.

To study the effect of peptide sequence and composition in LLPS, we selected FF derivatives with the general structure shown in Fig. 1a. We prepared a series of FF derivatives containing three or four amino acids: FFM, FFIba, FFF, FFFE-OMe, and MFF (Fig. 1/Supplementary Fig. 1/Supplementary Fig. 3). In addition to tripeptides, a dipeptide with a single F amino acid, LF, was synthesized and well characterized (Supplementary Figs. 4-17). All the FF peptides (20 mg mL$^{-1}$) were completely soluble in pH 6 buffer (5 mM HEPES, 100 mM NaCl) after a heating and cooling procedure, and the solutions remained

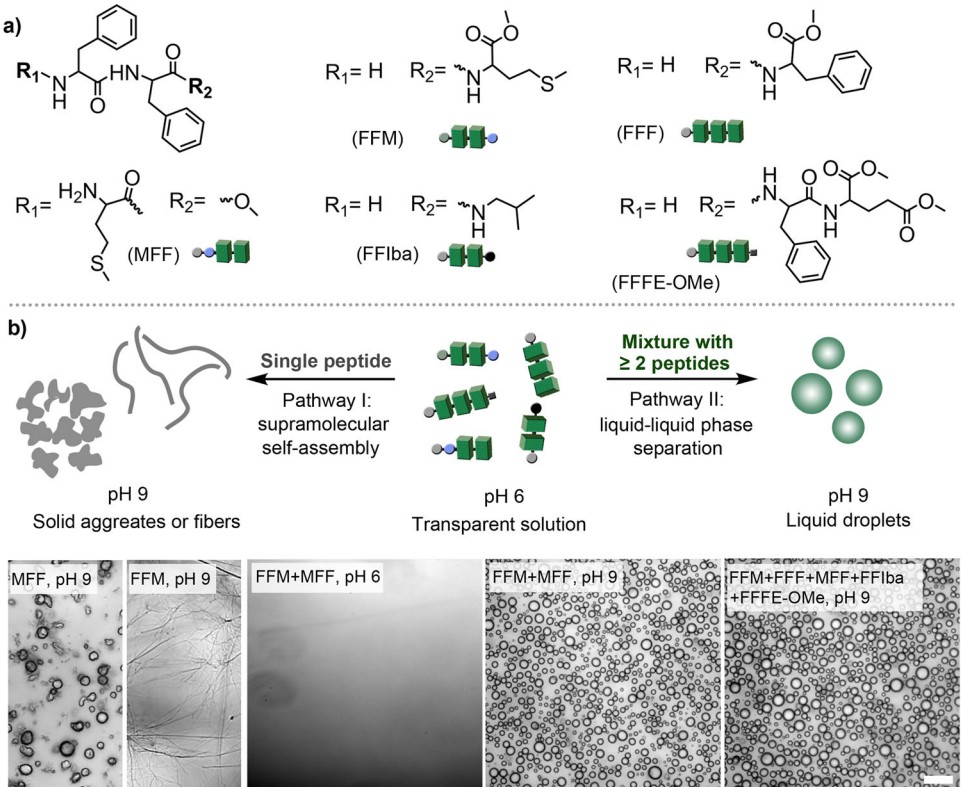

**Fig. 1 | Design and assembly of short peptide-based coacervates. a** Molecular structure of diphenylalanine-based short peptide molecules used in programmable phase separation; **b** Schematic of controlled and programmable phase separation. Pathway 1: Single peptides are soluble in acidic solutions and phase separate into solid irregular aggregates or fiber structures at pH 9. Pathway 2: A mixture of two or more short peptides resulted in the formation of liquid condensed droplets at pH 9, verified by bright field microscopy imaging, scale bar = 20 μm in all images. 3 experiments were repeated independently with similar results.

transparent after cooling down to room temperature. The phase separation behaviour of all peptides was dependent on the pH of the solution. For example, increasing the pH value to slightly basic levels (greater than 7.0) by adding a few drops of 0.1 M NaOH solution caused the FFM solution (5 mg mL$^{-1}$) to become significantly cloudy. Microscopic imaging analysis revealed the initial formation of liquid droplets which rapidly transformed into fiber-like structures in less than 2 min (Supplementary Figs. 18-19). Similar behaviour was observed with FFF, where the peptide was soluble at pH 6 and formed small coacervate droplets at pH 9, which then formed fibers after 10 min (Supplementary Fig. 20). The metastable formation of coacervates and their subsequent transformation into solid type aggregates within minutes was also observed with MFF, FFFE-OMe, and FFIba peptides (Supplementary Fig. 20 /21/23). The observed behaviour of the short peptides in the solution can be attributed to the charge of the -NH$_2$ group, which is positive at acidic pH and neutral at basic pH (pka of the α-amino group in F or M amino acid ~7). At low pH, electrostatic repulsion and increased hydration prevented the peptides from self-associating, keeping them water soluble. Conversely, at basic pH, the reduced solvation and lack of repulsive forces between the amino groups allow hydrophobic/aromatic interactions and van der Waals forces to induce phase separation, resulting in metastable coacervates. Over time, after the initial condensation, the peptides reorganized into an ordered packing arrangement. As a result, the coacervates rapidly transform into thermodynamically favourable microfibers or other types of solid aggregates, driven by π-π stacking and hydrogen bonding[42,53].

A completely different behaviour was observed when binary peptide mixtures were investigated. We found that coacervate droplets formed without transforming into fibers or solid aggregates when two or more short peptides were present in the solution. For example, a 1:1 mixture (weight ratio) of FFM and MFF, which differ only in the position of the methionine amino acid, formed a transparent solution at pH 6. When the pH was increased to approximately 9, microscopy images revealed the formation of relatively stable coacervate droplets without conversion to fibers or solid aggregates for at least 20 min of observation (Supplementary Fig. 24). The behaviour of the MFF: FFM mixture differs from that of solutions containing only MFF or FFM, which form unstable coacervates that rapidly transform into fibers and solid aggregates. To determine whether the higher stability of coacervates from binary peptide mixtures is a general property, we investigated different combinations and tested their phase separation behaviour. We observed the formation of stable liquid coacervates (lasting over 30 min) in each two-peptide 1:1 mixture (weight ratio) containing FFM, FFIba, FFF, FFFE-OMe, and MFF (Supplementary Figs. 25-33).

Coacervates with improved phase stability were also observed when mixing up to five peptides, allowing the formation of peptide coacervates with more complex compositions (Supplementary Fig. 34/35/36). The formation of liquid coacervates in mixtures of phase-separating peptides may be due to the disordered packing of peptide molecules in the condensed state, which is likely to inhibit their tendency to form solid aggregates such as fibers. To provide additional evidence, we mixed a phase-separating peptide, FFM or MFF, with a coacervate-forming peptide, FMF. FMF is soluble at pH 6 and forms relatively stable coacervate droplets at pH 9. When mixed with FFM or MFF, the mixtures also produced spherical droplets that did not transform into irregular aggregates or fibrous structures during observation (Supplementary Fig. 37). This result indicates that the stability of short-peptide coacervates is largely dependent on the compositional complexity of the coacervate phase. In the case of FFM, the complexity is achieved by simply rearranging the amino acids

within the short peptide sequences. However, the specific identity of the components also plays an important role, as demonstrated by mixing the phase-separating peptide FFM with the non-phase-separating peptide LF. LF is soluble at both pH 6 and pH 9, with no visible droplets or aggregates observed in the microscopic images (Supplementary Fig. 38). When FFM and LF were mixed, coacervates initially formed when the pH was raised to around 9, but these quickly (<5 min) transformed into a fibrous material, similar to the results obtained for single peptide solutions of FFM. In addition, when MFF peptide was mixed with other non-phase-separating peptides like MGG or MFG, a similar phenomenon was observed, i.e., initial coacervation and the formation of irregular aggregates (Supplementary Fig. 39/40). These results suggest that the nature of the second component would be important for the stabilization of coacervate droplets. In this case, LF, MGG, and MFG were highly water soluble, which limited their ability to form stable binary coacervates with FFM or MFF.

## Binary peptide coacervates (FFM and MFF)

After studying the general behaviour of solutions containing mixtures of short peptides, we selected FFM and MFF to further investigate the properties of coacervates obtained from these two peptides, which we will refer to as binary peptide coacervates (BPCs). Firstly, the effect of varying the weight ratio of FFM and MFF (from 9:1 to 1:9) was investigated. Their phase separation behaviours were recorded using microscopic imaging. The results showed that the FFM ratio had a significant effect on the resulting phase properties. For example, solutions with FFM weight ratios between 30% and 70% produced relatively stable liquid coacervates (Fig. 2a/Supplementary Fig. 24). In contrast, when the FFM weight ratio was below 30% or above 70%, i.e. when there was an excess of any of the two peptides, the initially formed binary peptide coacervates gradually transformed into either irregular solid aggregates or fibrous structures within 20 min of observation (Fig. 2a/Supplementary Fig. 24). Surprisingly, the BPCs

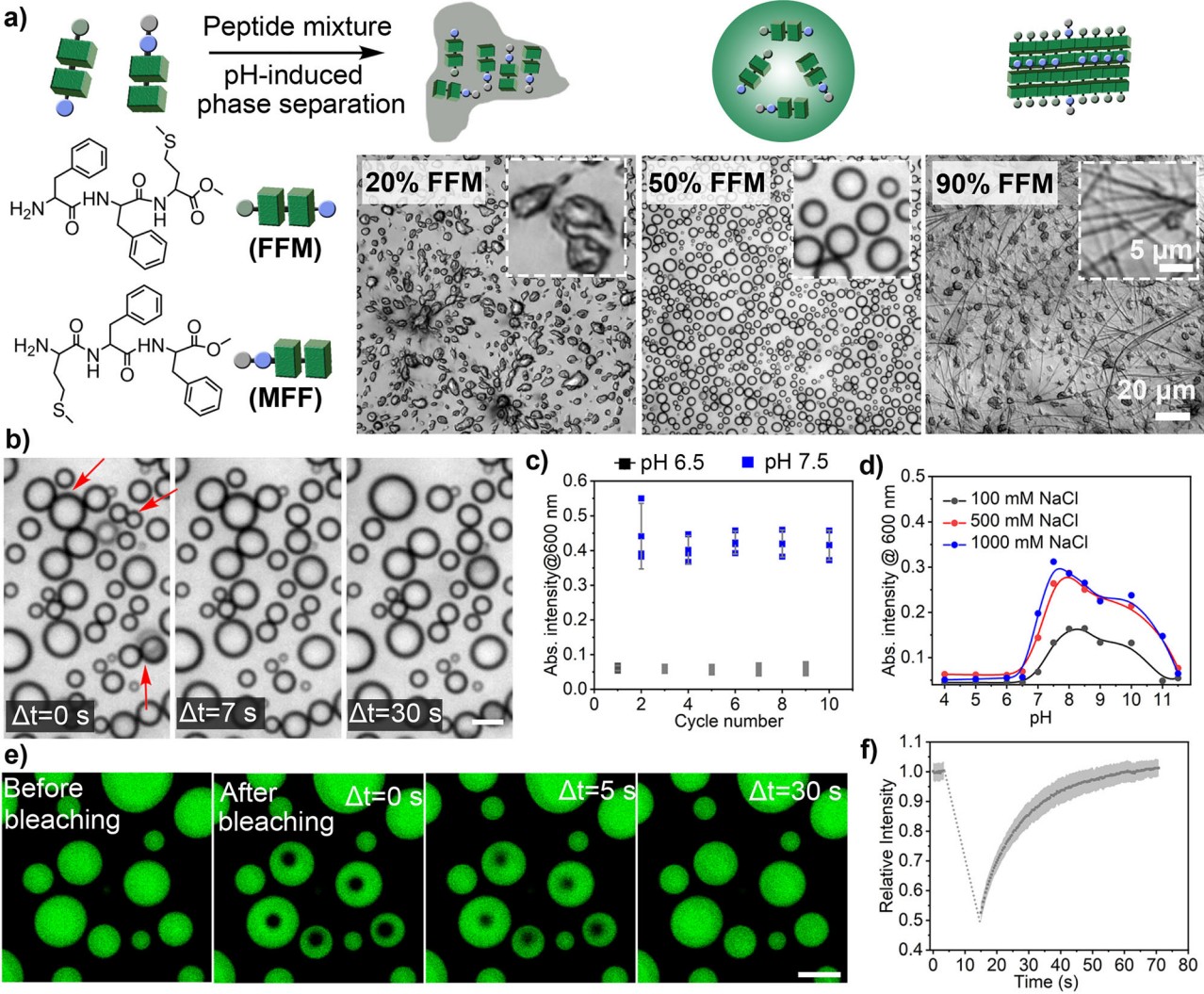

**Fig. 2 | Components, formation and key properties of peptide coacervates. a Top:** Schematic illustration of phase separation in binary peptide coacervates (FFM/MFF). **Bottom:** Microscopic imaging shows the formation of solid-type irregular aggregates or fibrous structures with 20% and 90% FFM (weight ratio), respectively. However, the use of 50% FFM in the peptide mixture (5 mg mL⁻¹ total) resulted in the formation of liquid condensates. Scale bar = 20 µm, inset scale bar = 5 µm. Three experiments were repeated independently with similar results. **b** The FFM/MFF condensates are liquid, as evidenced by their coalescence into larger droplets. Scale bar = 5 µm. Three experiments were repeated independently with similar results. **c** Turbidity of the FFM/MFF mixture (5 mg mL⁻¹ total) at pH 6.5 and pH 7.5, showing that the peptides can repeatedly transition between the solution state at pH 6.5 and the condensed state at pH 7.5 over several cycles. Data represent mean ± SD for $n = 3$ independent samples. Error bars depict the standard deviation (SD) obtained from microplate reader analysis. **d** Turbidity of FFM/MFF condensates (2.5 mg mL⁻¹ total) measured in buffers with different pH and NaCl concentrations. **e** Confocal images corresponding to FRAP in coacervates formed with the FFM/MFF mixture (1:1 weight ratio, 5 mg mL⁻¹ total) over time. Green emission is Rhodamine B. Scale bar = 5 µm. Three experiments were repeated independently with similar results. **f** FRAP traces of coacervates formed with the FFM/MFF mixture (1:1 weight ratio, 5 mg mL⁻¹ total) over time. Data represent mean ± SD for $n = 5$ representative microscopic images. Error bars (grey shaded area) depict the standard deviation (SD) from confocal imaging analysis.

prepared with a 50% FFM and 50% MFF ratio remained stable even after 3 days, with no significant liquid-to-solid transition observed (Supplementary Fig. 41). The exact ratio of FFM and MFF in the coacervate phase was verified by [1]H NMR, which indicated a proportion of FFM of 48.8% ± 0.9%, which is very close to the 1:1 feed ratio (Supplementary Fig. 42). The peptide content in the concentrated coacervate phase and the surrounding dilute phase was estimated using NMR with 1,3,5-trimethoxybenzene as an external standard. The results indicated that the peptide content ratio between the concentrated and diluted phases was approximately 1.25:1 (Supplementary Fig. 43). However, due to the substantial difference in volume between the bulk phase and the coacervate phase, the peptide concentration in the coacervate would be much higher than that in the surrounding diluted phase. Additionally, when mixtures of FFM and MFF were prepared with initial FFM proportions of 40% or 60% before coacervation, the calculated FFM ratios in the coacervate phase were ~40.5% and 58.3%, respectively, closely matching the initial feed ratios (Supplementary Fig. 44). Based on these results, a 1:1 weight ratio of MFF to FFM was used for the subsequent studies.

Microscopy revealed that BPCs at concentrations of 5 mg mL$^{-1}$ of FFM and MFF formed microdroplets, typically 1-10 μm (diameter), at pH ~9 (Fig. 2a). These peptide-rich condensates exhibited liquid-like properties as evidenced by their ability to coalesce into larger droplets (Fig. 2b). The relationship between fusion time (τ) and average droplet radius (l) for two coalescing droplets can be used to approximate the inverse capillary velocity (η/γ). In this model, the droplets are assumed to be dispersed in a low-viscosity medium, where τ ≈ l(η/γ). Here, η and γ represent the viscosity of the coacervate droplets and the interfacial tension, respectively[54]. For the peptide coacervate droplets, the η/γ value was determined to be ~ 0.044 s μm$^{-1}$, which is comparable to those of other peptide-based condensate or coacervate systems (Supplementary Fig. 45)[55]. This value is consistent with slow relaxation, indicative of soft and potentially viscoelastic droplets. The phase transition and droplet formation were sensitive to the peptide concentration and the pH of the solution (Fig. 2c/Supplementary Fig. 46/47). For example, droplet formation was observed when the total peptide concentration exceeded ~1 mg mL$^{-1}$ (Supplementary Fig. 46). Turbidity, as measured by UV-vis spectroscopy was highest from pH ~7.5 to ~10 in 100 mM NaCl buffer (Fig. 2d). The turbidity decreased significantly when the pH value was increased to ~11 or higher (Fig. 2d/Supplementary Fig. 48). When the NaCl concentration was increased to 500 mM and 1000 mM, there was a significant increase in turbidity of the peptide solution at neutral to alkaline pH, especially at pH ~7 and pH ~11 (Fig. 2d/Supplementary Fig. 49). This indicates that with increasing salt concentration, the inter/intra-molecular interactions between the peptide molecules are strengthened due to the enhanced hydrophobic effect promoted by the FF core. In addition, the electrostatic repulsion between the peptides is screened by the electrolyte, favouring phase separation.

The binary peptide coacervates exhibited high resistance to electrolytes, in contrast to polyelectrolytes-based complex coacervates, which typically dissolve in the presence of >500 mM NaCl[7]. The phase transition between the solution and coacervate states in the binary peptide mixture is highly reversible and can be repeated over several cycles between pH ~6.5 and ~7.5 (Fig. 2c/Supplementary Fig. 50). While conventional coacervate droplets formed from synthetic polymers, RNA, and proteins typically have a molar mass greater than 10 kDa, the peptides in BPCs are much smaller (<0.5 kDa), contributing to their dynamic properties[8]. The molecular mobility of the peptides in BPCs was investigated using fluorescence recovery after photobleaching (FRAP). BPCs formed from a mixture of FFM and MFF showed almost complete fluorescence recovery approximately 60 sec after photobleaching, confirming their highly dynamic nature and liquid-like properties (Fig. 2e). Furthermore, even after incubation for up to 3 days, the droplets exhibited a level of fluorescence recovery

comparable to that of fresh droplets, indicating no significant changes in molecular mobility or liquid-like properties over extended incubation periods (Supplementary Fig. 51). In contrast, the extent of fluorescence recovery decreased significantly in aggregates containing 100% MFF (Supplementary Fig. 52). This decrease in the mobile fraction is consistent with stronger, undisturbed intermolecular interactions between MFF when it is present in excess. In BPCs, these interactions are diluted and disrupted by the presence of FFM, resulting in stable droplets with liquid-like properties.

To investigate the role of hydrophobic interactions in phase separation, 1,6-hexanediol, a compound known to disrupt weak hydrophobic protein-protein interactions, was added to the coacervate mixture[56]. The turbidity obtained in the presence of ~0.8 M 1,6-hexanediol was reduced to approximately 75% of the turbidity of the solution before the addition of 1,6 hexanediol (Supplementary Fig. 53). However, microscopic images showed that droplet formation persisted in the presence of ~0.8 M 1,6-hexanediol (Supplementary Fig. 53) despite the reduction in turbidity. These results indicated that hydrophobic interactions play a role in droplet formation, but are not the only driving force. Other possible interactions include hydrogen bonding. The role of hydrogen bonding in driving the phase transition was investigated by treating the FFM/MFF coacervates with urea, a compound known to disrupt non-covalent polar interactions in aqueous solutions[57]. Increasing the concentration of urea (up to 5 M) resulted in a sharp decrease in the turbidity (at $\lambda_{abs} = 600$ nm) of the FFM/MFF droplet solution (5 mg mL$^{-1}$, Supplementary Fig. 54). Microscopic imaging revealed a significant decrease in the number density of the peptide condensates when treated with 0.5 M urea. When the urea concentration was further increased to 5 M, most of the droplets dissolved (Supplementary Fig. 54). These observations suggested that hydrogen bonding plays an important role in driving liquid-liquid phase separation and condensate formation. The more pronounced effect of urea compared to that of 1,6-hexanediol on the number density of BPCs indicates that hydrogen bonding is a primary driving force for droplet formation.

To further elucidate the molecular mechanisms that cause the difference in phase behaviour between single peptide and binary peptide coacervates, we then performed all-atom molecular dynamics (MD) simulations. Starting from the initial fully dissolved state at a concentration of 80 mg mL$^{-1}$, the MD simulation showed dense aggregates in an aqueous solution containing only MFF or FFM peptide. In contrast, simulations involving the MFF/FFM mixture resulted in relatively loose aggregates (Supplementary Fig. 55), consistent with the more mobile phase observed experimentally for BPCs. When the peptide concentration was doubled to 160 mg mL$^{-1}$, the simulation showed that MFF or FFM aggregates transformed into fiber-like structures (Fig. 3a/Supplementary Fig. 55). In contrast, the cluster obtained from a mixture of MFF and FFM remained loose and showed no evidence of extensive fiber-like nucleation (Fig. 3a/Supplementary Fig. 55). To quantitatively differentiate the fiber-like precursors from the condensates, we evaluated the aggregation propensity and clustering degree from the MD simulation[58,59]. Aggregation propensity (AP) can be measured by comparing the solvent-accessible surface area (SASA) of peptide molecules from their initial to final simulated states. This method effectively evaluates assembly potential, including fiber formation[58]. Similarly, the clustering degree (CD), expressed as the ratio of the largest cluster size to the total number of peptide molecules in the system, serves as a reliable predictor of condensate formation[59]. Additionally, hydrogen bonding (H-bond), characterized by a donor-acceptor angle of less than 30 degrees and a donor-acceptor distance of less than 0.35 nm, is a key indicator of fibril formation[50]. Based on these parameters, it can be concluded that the FFM + MFF system exhibits a lower degree of clustering and reduced aggregation propensity compared to pure MFF or pure FFM systems. This is consistent with a relatively poor fiber-forming ability. (Fig. 3b/Supplementary Fig. 55). Based on the coarse-grained MD simulations

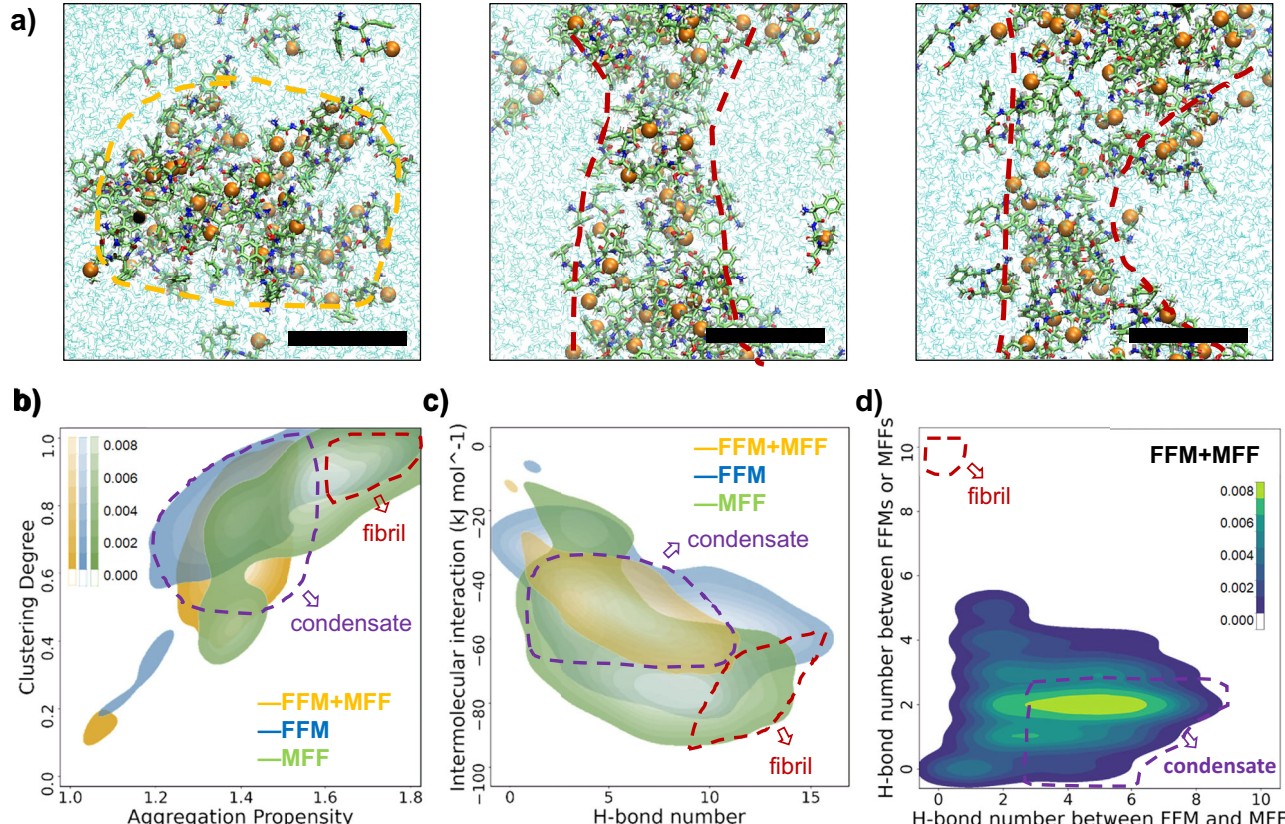

**Fig. 3 | Molecular dynamics simulation of the aggregation states of the FFM/MFF mixture, pure FFM, and pure MFF. a** Snapshot of FFM + MFF, pure FFM, and pure MFF with a concentration of 160 mg mL⁻¹. **b** The clustering degree of pure FFM, pure MFF and FFM + MFF mixture plotted as a function of their aggregation propensity values from the MD simulation. **c** The intermolecular interaction values of pure FFM, pure MFF and FFM + MFF mixture plotted as a function of their intermolecular H-bond numbers from the MD simulation; **d** A comparison of H-bond number between FFM + MFF, and among FFMs or MFFs in the peptide mixture system. Scale bars: 2 nm.

of dipeptides performed by Tang et al, the fiber formation is characterized by AP > 3 and CD > 0.9, while condensate formation is defined by 1 < AP < 3 and CD > 0.5[59]. However, in our previous all-atom MD simulations of z-FF peptides, the boundary between condensates and fibers shifted to lower AP values. In these simulations, condensates were stable when the SASA ranged from 0.65 to 0.85 (equivalent to 1.2 < AP < 1.6), and fiber formation occurred when the SASA dropped below 0.65 (equivalent to AP > 1.6)[50]. Therefore, in this study, we propose revised definitions: fibers are defined as AP > 1.6 and CD > 0.9, while condensates are defined as 1.2 < AP < 1.6 and CD > 0.5. Intermolecular hydrogen bonding (H-bonding) analysis revealed that the number of hydrogen bonds formed in the MFF + FFM mixture was lower than in the pure MFF or pure FFM system, indicating a weaker intermolecular interaction between the peptide molecules in the mixed system (Fig. 3c/Supplementary Fig. 56). Notably, while single peptide systems form fibers, they also occupy regions in the simulated data associated with condensate formation (Fig. 3b, c). This suggests that the single peptide systems are capable of forming condensates at an early stage prior to the irreversible formation of fibers. Fiber formation occurs only when the system transitions to regions with high values of AP, CD and H-bond number (red outlines in Fig. 3b, c). In addition, in the mixed peptide system, the number of hydrogen bonds formed between FFM and MFF was greater than those between FFMs or MFFs alone (purple region in Fig. 3d). This increased interaction between the two peptides hinders the nucleation and growth of nanofibril composed of pure FFMs or MFFs within the formed condensates (red region in Fig. 3d). These simulation results confirmed from a microscopic perspective that the mixed system has a lower propensity for aggregation and clustering, fewer intermolecular H-

bonds, and weaker intermolecular interactions, leading to the occurrence of liquid-liquid phase separation.

## Multi-responsiveness of the peptide coacervates

The amino group in the peptide molecules allows direct control of coacervate formation by changing the pH value of the environment. In addition to pH responsiveness, the peptide coacervates also exhibited dynamic behaviour in response to various triggers. For example, when subjected to multiple heating (80 °C) and cooling (to room temperature) cycles, BPCs (composed of FFM/MFF, total 5 mg mL⁻¹) in 5 mM HEPES buffer at pH 8 showed high reversibility with no evidence of irreversible aggregation. Coacervation and dissolution were observed with each thermal cycle (Fig. 4a-c/Supplementary Fig. 57). This was confirmed by turbidity tests performed during the heating and cooling cycles (Fig. 4b). Microscopic imaging further confirmed coacervation in the initial phase or after cooling, as well as dissolution upon heating (Fig. 4c/Supplementary Fig. 58). The observed temperature responsiveness of FFM/MFF binary peptide coacervates (BPCs) is due to heating, which increases the solvation level of the peptide molecules and disrupts the numerous weak molecular interactions that hold the droplets together, leading to the dissolution of the peptide molecules in the buffer. Upon cooling, the peptide molecules desolvated, promoting phase separation. The turbidity after cooling did not revert to its initial level, possibly due to variations in the coacervation rate, which influenced both the number and size of the reformed droplets. Coacervates formed almost instantly when the pH of the acidic peptide solution was raised to about 8. However, after the heating and cooling cycle, the process slowed down, likely because the coacervates settled and stuck to the walls of the measurement chamber.

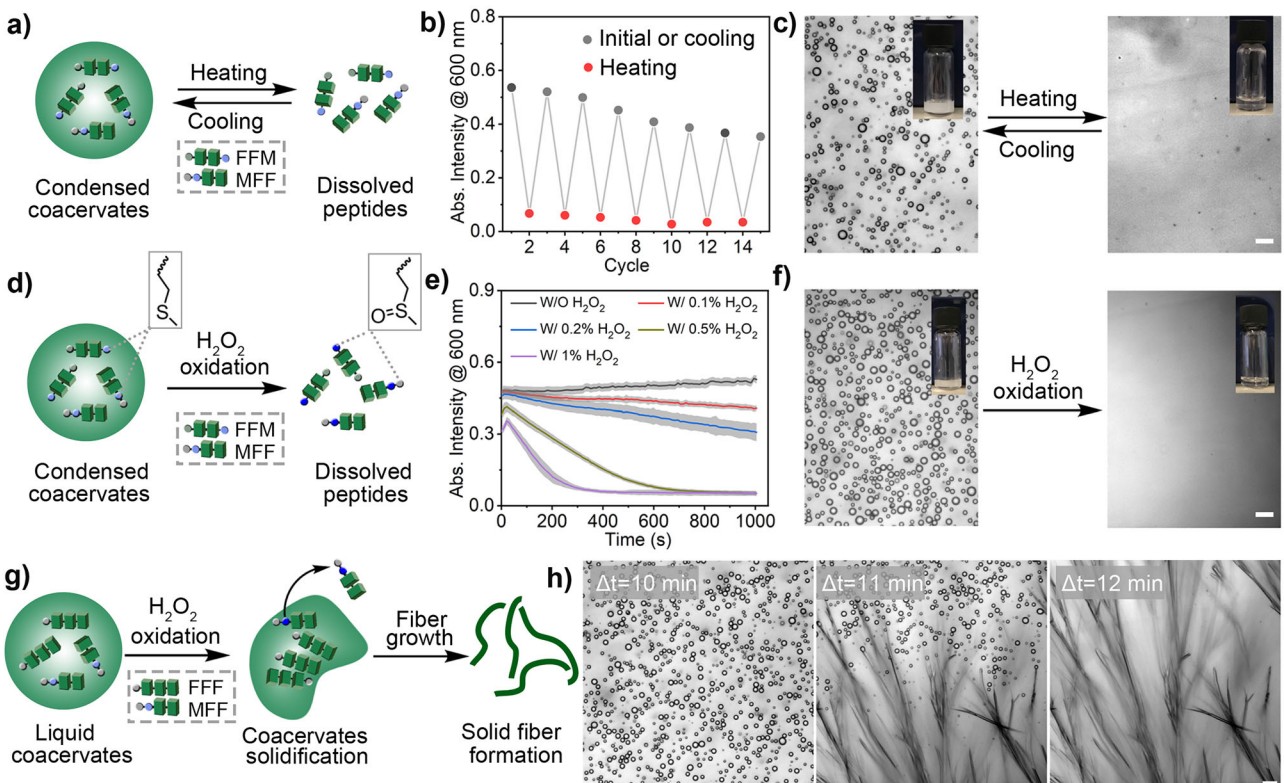

**Fig. 4 | Multi-responsive behaviour of peptide coacervates under different external stimuli. a** Thermo-responsiveness: Schematic representation of the thermo-responsiveness of peptide coacervates. The peptide condensates dissolve when heated but can reform when cooled. **b** Turbidity changes: The turbidity of FFM/MFF peptide condensates (5 mg mL⁻¹) was monitored upon heating (80 °C for ~1 minute) and subsequent cooling (room temperature water bath for ~2 min). **c** Microscopic imaging: images of FFM/MFF peptide condensates (5 mg mL⁻¹) before and after heating at 80 °C for ~1 minute. Scale bar = 5 μm. Three experiments were independently repeated with similar results. **d** Oxidation sensitivity: Schematic illustration of the oxidation-responsive behaviour of peptide coacervates. The peptide condensates gradually dissolve in the presence of H₂O₂ due to the oxidation of thioether groups, which increases the hydrophilicity of the peptide molecules. **e** Turbidity changes with H₂O₂: Turbidity changes of FFM/MFF peptide condensates (5 mg mL⁻¹) in the presence of different concentrations of H₂O₂. Data

represent mean ± SD for $n = 3$ independent samples. Error bars (grey shaded area) represent the standard deviation ($n = 3$) from microplate reader analysis. **f** Microscopic images after H₂O₂ treatment: Microscopic images of FFM/MFF peptide condensates (5 mg mL⁻¹) after treatment with 1% H₂O₂ for 10 min. Scale bar = 5 μm. Three experiments were independently repeated with similar results. **g** Transition to solid fibrous structure: Schematic illustration of the transition from liquid condensates to a solid fibrous structure with a mixture of FFM and FFF molecules (1:1 weight ratio), utilizing the oxidation-responsiveness of the FMM peptide. **h** Microscopic imaging of the solid-state transition: Microscopic images of FFF/MFF peptide condensates (5 mg mL⁻¹) showing a gradual solid transition upon treatment with 1% H₂O₂ over ~11 min, which completely transforms into a fibrous structure in ~12 min. Scale bar = 10 μm. Three experiments were independently repeated with similar results.

The BPCs are sensitive to organic solvents such as acetonitrile. When mixed with 20% volume of acetonitrile, the droplets disintegrated, resulting in a clear solution. This was confirmed by turbidity tests and microscopic imaging (Supplementary Fig. 59). As the acetonitrile evaporates from a peptide solution initially containing 50% volume of acetonitrile, small condensates form. These small condensates gradually coalesce into larger droplets (Supplementary Fig. 59). This phenomenon can be attributed to the decreasing level of acetonitrile, which restores the inter/intra-molecular interactions of the peptides and drives the phase separation into condensates[60]. As a control, coverslips were used to inhibit solvent evaporation. Under these conditions, the peptide solution treated with 50% volume acetonitrile remained transparent and did not form droplets even after prolonged observation (Supplementary Fig. 59).

The presence of methionine in the peptide allows the control of coacervate formation by oxidation chemistry under mild conditions[31]. First, the H₂O₂-responsive nature of the peptides (using MFF as an example) was confirmed by ¹H NMR. After 15 min of incubation with 0.5% H₂O₂, approximately 40% of MFF (5 mg mL⁻¹ in D₂O) was oxidized. The oxidation level exceeded 90% after 80 min of incubation, demonstrating the effective ROS-responsive property of the peptide (Supplementary Fig. 60). Studies have demonstrated that converting thioether groups

into more polar sulfoxide or sulfone forms enhances the water solubility of poly(L-cysteine) and poly(L-methionine) derivatives, causing peptide-based structures to disassemble[61]. Here we show that dynamic disassembly controlled by redox chemistry can also be achieved with BPCs. To actively regulate coacervation in FFM/MFF, the responsiveness of thioether groups was studied to mimic the post-translational modifications that control protein coacervation in cells. (Fig. 4d)[23]. Upon oxidation of the methionine residue in the FFM/MFF peptide mixture with H₂O₂, the hydrophobic thioether group is converted to hydrophilic sulfone or sulfoxide species. This increased hydrophilicity of the oxidized FFM/MFF peptides in BPCs results in the dissolution of the coacervate phase due to the higher degree of solvation of the peptides. This process was evidenced by a significant decrease in turbidity of the BPC solution (FFM/MFF, 5 mg mL⁻¹ in 5 mM HEPES buffer) after treatment with different concentrations of H₂O₂. The initial cloudy dispersion of BPCs became a clear solution (Fig. 4e, f). Microscopic imaging shows that the number of initial BPC droplets gradually decreased and disappeared after treatment with 1% H₂O₂, transforming into a clear solution within 5 min of incubation (Supplementary Fig. 61). By exploiting the oxidative reactivity of methionine, we were able to achieve a controlled transition from liquid BPCs to fibers. This was achieved by producing BPCs containing FFF and MFF (Fig. 4g). The tripeptide FFF (C-

terminal methyl ester) rapidly forms fibers when it is the only component in the solution. MFF was used as the responsive component whose ability to form coacervates can be controlled by oxidation. Initially, BPC droplets were formed by raising the pH of an acidic solution of the peptide mixture (FFF/MFF in a 1:1 weight ratio, with a total concentration of 5 mg mL$^{-1}$) to approximately 8 (Fig. 4h). These condensed droplets remained relatively stable, with no solid transition observed during 1 h of observation (Supplementary Fig. 62). Treatment with 1% H$_2$O$_2$ did not affect the initial formation of BPCs. However, after approximately 11 min of incubation, a sharp solid transition and fiber growth were observed, with all droplets transforming into fibrous structures within minutes (Fig. 4h/Supplementary Fig. 63). This phenomenon was attributed to the oxidation of the thioether in the MFF molecule, increasing its solvation level. The oxidized MFF gradually dissolved in the buffer, resulting in a reduced proportion of MFF in the coacervate phase and an increased proportion of FFF. This resulted in an increasingly ordered packing of FFF and facilitated an irreversible liquid-to-solid phase transition.

### The partitioning of molecules in BPCs

Having demonstrated the formation and responsiveness of BPCs, we investigated their potential for partitioning and concentrating guest cargoes. Rhodamine isothiocyanate-labelled BSA (RITC-BSA) was used as a model protein, along with other functional enzymes such as Cy5-modified glucose oxidase (Cy5-GOX) and rhodamine isothiocyanate-conjugated horseradish peroxidase (RITC-HRP). By simply mixing these proteins and enzymes with the BPCs (FFM/MFF, 1:1, 5 mg mL$^{-1}$ total), they were sequestered and distributed within the coacervate droplets with partition coefficients of 56, 44, and 48, respectively (Supplementary Fig. 64). The sequestration and confinement of the enzymes within the peptide coacervates allowed their use as biomimetic reactors. To further investigate their catalytic efficiency, FFM/MFF BPCs containing HRP (2.5 µg/mL) were treated with 0.2 µL of Amplex Red substrate in DMSO (1 mM) and 0.01% H$_2$O$_2$ (volume). Confocal imaging of the reaction showed that the fluorescent resorufin product ($\lambda_{em}$ ~ 580 nm) was mainly confined within the coacervates (Supplementary Fig. 65).

In addition to their ability to concentrate biomacromolecules like polyelectrolytes-based complex coacervates, binary peptide coacervate droplets can also sequester a variety of small guest molecules. We evaluated the ability of BPCs (FFM/MFF, 5 mg mL$^{-1}$ total) to concentrate molecular cargoes using confocal imaging (Supplementary Fig. 66). As shown in Supplementary Fig. 67, aromatic fluorophores with neutral or positive charges, such as Nile Red, thioflavin T (THT), methylene blue, rhodamine B, and rhodamine 6 G, were all concentrated within the BPCs at pH ~8, with apparent partition coefficients (K = F$_{droplet}$/F$_{background}$) of 781, 772, 169, 185, and 48, respectively. Variations in K were due to different affinities for the microenvironments provided by BPCs (Fig. 5b-d). Negatively charged dyes with a hydrophobic character, such as resorufin, were also internalized and concentrated inside the peptide coacervates with an apparent partition coefficient of 46 (Fig. 5g). However, a dramatic decrease in K was observed in cargoes with increasing negative charge, such as for sulforhodamine B (Fig. 5e). With a further increase in hydrophilicity, as seen for rhodamine 110, calcein, and pyranine, the partition coefficients decreased even further to 2.4, 0.5, and 0.5, respectively (Fig. 5f/Supplementary Fig. 67). These results indicate that the binary peptide coacervates (BPCs) have a higher propensity to sequester aromatic species and cationic cargoes with hydrophobic affinity. The partitioning of these compounds is facilitated by π-π and cation-π interactions between the guest molecules and the phenylalanine residues within the droplets[62]. This suggests that the microenvironment within the BPCs allows them to accommodate a wide variety of guest molecules, effectively acting as a concentrating host for species with hydrophobic affinity. For instance, as shown in Fig. 5h/Supplementary Fig. 68, Nile Red exhibited very weak emission when dispersed in PBS. However, its emission was significantly stronger when dissolved in CH$_3$CN or encapsulated in the BPCs. The emission intensity of Nile Red within BPCs was approximately 244 times greater than in PBS. This enhanced emission intensity can be attributed to the sequestration and concentration of the hydrophobic dye within the droplets.

### Designing cellular microreactors with BPCs

Having shown that BPCs can concentrate hydrophobic species, we investigated their potential as microreactors to facilitate the synthesis of complex molecules. Transition metal catalysts (TMCs), known for their high efficiency and chemical versatility, are excellent candidates for bioorthogonal catalysis, enabling specific chemical reactions within living systems[46]. Due to low water solubility and limited biocompatibility, maintaining the activity of TMCs often requires incorporating them into nanoscale supports or polymer-based scaffolds[63–65]. Recently, we have shown that single peptide coacervates can sequester active TMCs, resulting in microreactors[37]. The integration of hydrophobic TMCs into microscale coacervate-based liquid droplets provides a versatile method for designing cell-like systems with applications in bio-inspired catalysis. We explored the use of FFM/MFF BPCs as compartments to create TMC microreactors. Ruthenium-catalyzed deallylation was chosen to evaluate the performance of the microreactors. The reaction involves the reactivation of the fluorescence of an allylcarbamate caged pro-fluorophore in solution and living cells. The microreactors were prepared by sequestration of [Cp*Ru(cod)Cl] (Cp = pentamethylcyclopentadienyl, cod = 1,5-cyclooctadiene, abbreviated as Ru) inside the BPCs by simple mixing during pH-induced condensate formation (Fig. 6a).

The catalytic activity of Ru-integrated BPC microreactors was investigated using alloc-protected rhodamine 110 (Rho-pro, Supplementary Fig. 2) as a substrate. As a result of the reaction, the fluorescence intensity ($\lambda_{em}$ = 525 nm) gradually increased in the solution (5 mM HEPES, 100 mM NaCl, 1% DMSO) of Rho-pro (0.05 mg mL$^{-1}$) in the presence of Ru-BPC microreactors (Ru 0.05 mg mL$^{-1}$, peptides 5 mg mL$^{-1}$). After 20 min of incubation, we observed bright green fluorescence emission from the decaging of rhodamine 110 (Fig. 6b, c, Supplementary Fig. 69). The presence of the BPC droplets was essential for the success of the reaction. Reactions with the non-encapsulated Ru catalyst resulted in a negligible increase in fluorescence intensity (Fig. 6b, c). The Ru catalyst is inherently insoluble in aqueous solutions and, without a suitable carrier, is expected to exhibit reduced activity in an aqueous environment. The fluorescence intensity profile in the presence of coacervates showed a linear fit with an apparent rate constant of $0.789 \pm 0.003 \, s^{-1}$ compared to $0.0479 \pm 0.0007 \, s^{-1}$ for the reaction without coacervates at the same pH of ~9 (Fig. 6b). The presence of peptide coacervates resulted in a 16-fold increase in reaction rate. This increase can be attributed to the hydrophobic microenvironment within the peptide coacervates, which can concentrate reagents and maintain the activity of the Ru catalyst similar to organic solvents or polymeric supports.

To further demonstrate the potential of BPCs in promoting the synthesis of complex compounds, we explored their application in the copper-catalyzed alkyne-azide cycloaddition (CuAAC) reaction (Fig. 6d). CuAAC is a widely used bioorthogonal reaction and a versatile tool in organic synthesis, bioconjugation, and particle/cell surface functionalization[66]. It has broad applications in medicinal chemistry, chemical biology, and materials science[67]. Due to the inherent incompatibility of copper catalysts (and most substrates) with aqueous environments, catalysts are often immobilized on solid supports such as silica particles and metal-organic frameworks or conjugated with polymeric structures to increase their compatibility with water[68,69]. We investigated the potential of binary peptide coacervates in the CuAAC reaction. Cu-BPC microreactors were prepared by sequestering the bromotris(triphenylphosphine)copper(I) [CuBr(PPh$_3$)$_3$] catalyst in BPCs. The reagents 3-Azido-7-hydroxycoumarin, a pro-fluorophore, and phenylacetylene were selected to generate the fluorescent triazole

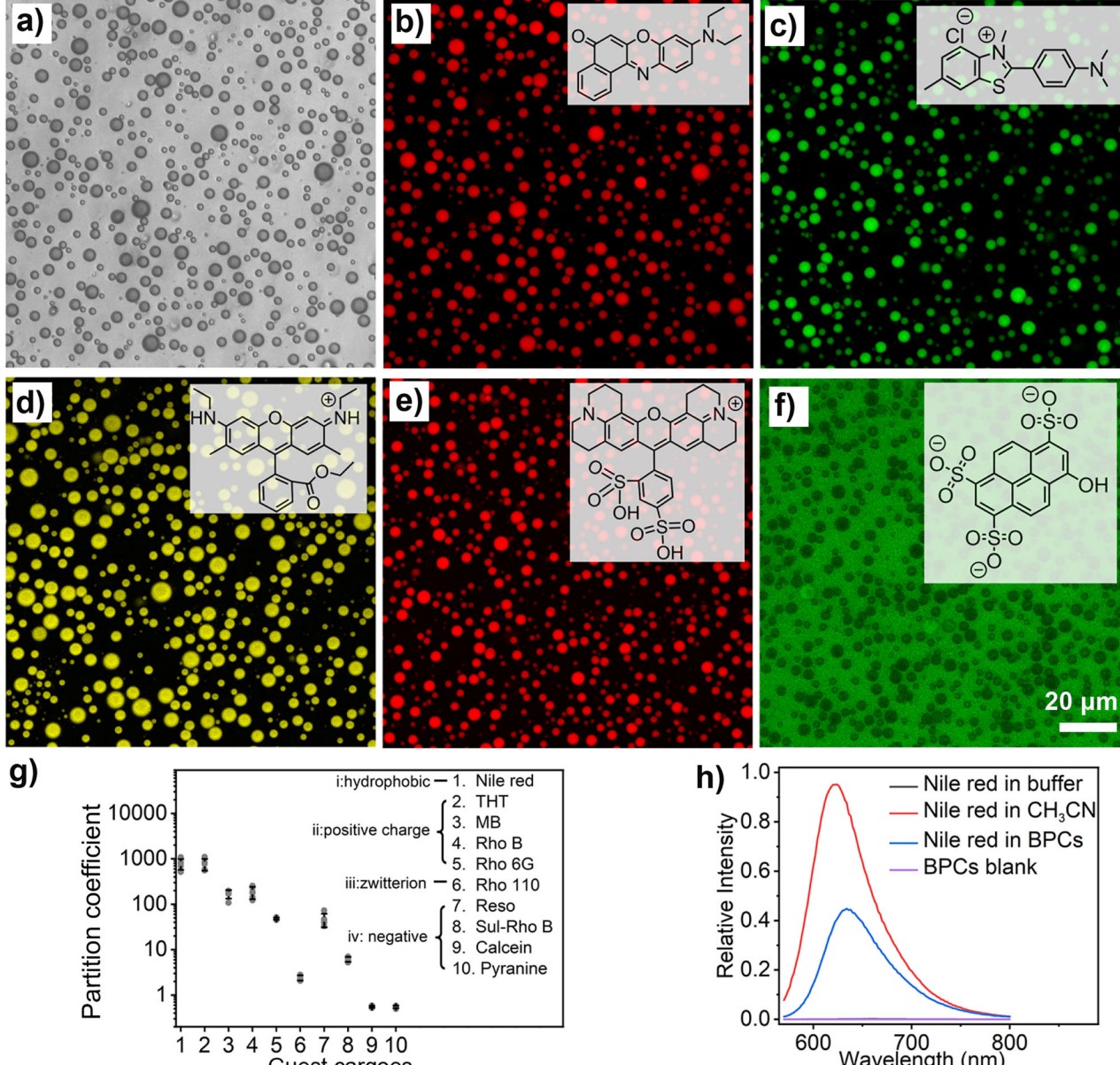

**Fig. 5 | Partitioning of guest molecules in binary FFM/MFF peptide condensates.** Confocal images of FFM/MFF binary peptide condensates (1:1 weight ratio, total 2.5 mg mL⁻¹) after incubation with different guest molecules: **a** bright field with Nile Red; **b** Nile Red; **c** Thioflavin T (THT); **d** Rhodamine 6 G; **e** Sulforhodamine B; **f** Pyranine. Scale bar: 20 µm. Three experiments were repeated independently with similar results. **g** Partitioning of guest molecules in BPCs. The partitioning of different guest molecules was determined by confocal imaging. The partition coefficient of the guest molecules was compared by analyzing the fluorescence intensity within the coacervates and in the surrounding environment. Data represent mean ± SD for $n = 5$ representative microscopic images. Error bars depict the standard deviation (SD) from confocal imaging analysis. **h** Fluorescence emission curve of Nile Red (0.05 mg mL⁻¹) in HEPES buffer, acetonitrile, binary peptide coacervates (BPCs), and blank BPCs.

complex by the click reaction mediated by Cu-BPC microreactors. UV-vis spectroscopy showed that the fluorescence intensity of the product ($\lambda_{em} = 450$ nm) gradually increased in solution (5 mM HEPES, 100 mM NaCl, 1.5% DMSO) in the presence of Cu-BPC microreactors (Cu 0.05 mg mL⁻¹, peptides 5 mg mL⁻¹). After 60 min of incubation with Cu-BPC microreactors, we observed bright blue fluorescence emission indicating the successful product formation (Fig. 6e, f). Like the results obtained for Ru-BPC microreactors, the click reaction occurred effectively only under conditions where the Cu catalyst was sequestered by the BPCs.

The fluorescence intensity in the presence of Cu-BPC microreactors was ~17-fold higher compared to the reactions carried out in the presence of the catalyst, but in the absence of the BPCs (Fig. 6f/

Supplementary Fig. 70). These results were attributed to the inherent hydrophobic nature of both the catalyst and the reagents, which without a suitable host exhibit reduced activity in an aqueous environment. The binary peptide coacervates, due to their hydrophobic internal microenvironment, can concentrate and partition the hydrophobic catalyst and substrates, facilitating the cycloaddition reaction. This enhanced catalytic effect observed with peptide-based coacervates is similar to polymeric particles with compartmentalized hydrophobic interiors that can isolate catalytic species in the hydrophobic domain. Compared to polymeric structures that require complex synthetic processes, bio-inspired peptide-based coacervates offer a promising alternative paradigm for the design of microreactors.

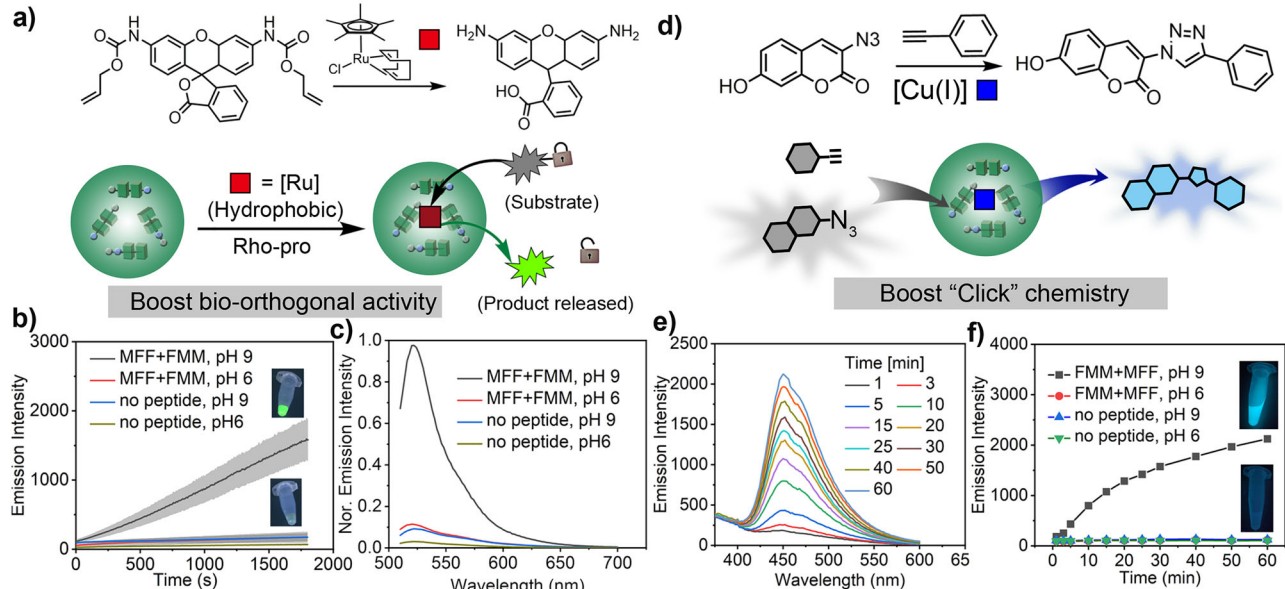

**Fig. 6 | Catalytic properties of binary peptide coacervates. a** Schematic representation of Ru-BPC microreactors and the Ru-catalysed uncaging of a pro-fluorophore. The reaction results in the release of a highly emissive fluorophore. **b** Fluorescence emission from uncaging reactions. Upon incubation in the presence of Ru-BPC microreactors (0.05 mg mL⁻¹ Ru, 5 mg mL⁻¹ peptides), there is a significant increase in fluorescence emission ($\lambda_{em}$ = 525 nm). In contrast, only a very slight increase in emission was observed in the absence of Ru-BPC microreactors. Conditions where BPCs are present: peptides (FFM/MFF) and pH 9. Conditions without BPCs present: peptides and pH 6 or no peptides. Data represent mean ± SD for $n$ = 3 independent samples; Error bars (grey shaded area) represent the

standard deviation ($n$ = 3) from microplate reader analysis. **c** Reaction progress after 20 min. The fluorescence emission intensity is significantly higher for reactions performed in the presence of Ru-BPC microreactors. **d** Schematic representation of Cu-BPC microreactors used in copper-mediated click chemistry. The Cu-BPCs contain the CuBr(PPh₃)₄ catalyst. The reaction when the reagents, azido-7-hydroxycoumarin, and phenylacetylene substrates, are incubated with the Cu-BPC microreactors. **e** Fluorescence emission curves of the formed product at different times during the click-reaction in the presence of Cu-BPC microreactors (FFM/MFF peptides, 5 mg mL⁻¹, pH 8). **f** Fluorescence emission intensity ($\lambda_{em}$ = 451 nm) of the click reaction product in the presence/absence of Cu-BPC microreactors.

## Construction of multi-compartment logic gates

Living cells are typically multi-compartmental structures with a hierarchical organization, containing a lipid bilayer membrane and various sub-organelles[3]. Currently, the design of biomimetic multi-compartment systems can achieved by incorporating complex coacervates into polymersomes or liposomes. However, these do not fully mimic the condensed and concentrated environment of the cytosol in living cells[70,71]. Having demonstrated that binary peptide coacervates can function as biomimetic reactors, we explored their use as models of biomolecular condensates (e.g. membraneless organelles) in the construction of multicompartment artificial cells (MACs).

Multicompartment artificial cells (MACs) were prepared by integrating binary peptide coacervates as internal compartments within complex coacervates formed from positively charged quaternised amylose (Q-Am) and negatively charged carboxymethylated amylose (C-Am) (Supplementary Fig. 71). To enhance the colloidal stability of the MACs, membranisation was achieved by interfacial assembly of BSA-modified MnO₂ nanoparticles (BSA@MnO₂) (Supplementary Fig. 72). Confocal imaging revealed the successful formation of MACs. This was verified by selectively labeling the BPCs and the complex coacervate phase with Nile Red and FITC-BSA, respectively (Fig. 7a, b). The fluorescence emission of Nile Red was exclusively localized within the BPCs, whereas FITC-BSA was enriched in the complex coacervate phase. This distribution occurs because the highly charged complex coacervates preferentially partition hydrophilic proteins and enzymes, while the microenvironment within the binary peptide coacervates more effectively sequester hydrophobic species.

The hierarchical organization and confinement of BPCs within the complex coacervates were further confirmed by confocal 3D reconstruction (Fig. 7b/Supplementary Fig. 73). The responsiveness of BPCs to chemical inputs and the effect of BPCs on catalysis allowed the construction of multi-compartment artificial cells with OR and AND

logic gate properties. To prepare an OR-MAC, we explored the responsiveness of BPCs to HCl (H⁺, input 1) and H₂O₂ (input 2) (Fig. 7c, d). Both inputs produce the same result, the disassembly of the internal BPC droplet (output). The disassembly was caused by the increase in peptide hydrophilicity through the protonation of the amino group of FFM/MFF (input 1) or the oxidation of the methionine (input 2). The OR-MACs, therefore, produce an output in the presence of either input or their combination. (Fig. 7g, h/Supplementary Fig. 74/75/76). In addition, MACs with AND functionality can be constructed by exploiting the fact that the Ru catalyst [Cp*Ru(cod) Cl] exhibits activity in aqueous solutions only when sequestered in binary peptide coacervates (BPCs). Therefore, AND-MACs were created by considering the presence of BPCs and the Ru catalyst as the two inputs and the catalytic reaction as the output. The uncaging reaction of Rho-pro was used to evaluate the response of the AND-MACs. As expected, the reaction (output) only occurs when both inputs are present (Fig. 7e, f, i, j Supplementary Fig. 77). In the absence of either the Ru catalyst or the BPC, no product formation was observed in AND-MACs (Supplementary Fig. 78/79). Our results showed that the integration of BPCs extended the performance of conventional biomimetic systems to include strategies for controlling reaction pathways and internal organization. This approach could further extend the construction of complex biomimetic systems with controlled dynamics and catalytic functions that closely resemble living systems.

In summary, we have presented the construction of dynamic coacervate droplets as an attractive model for biomolecular condensates by programming the liquid-liquid phase separation of binary short peptides. The extension from conventional single peptide assembly to binary or higher combinations of peptides results in the formation of stable coacervate droplets instead of fibers and solid aggregates. This is a step towards mimicking cellular phase separation behaviour with multi-components and advances the field of

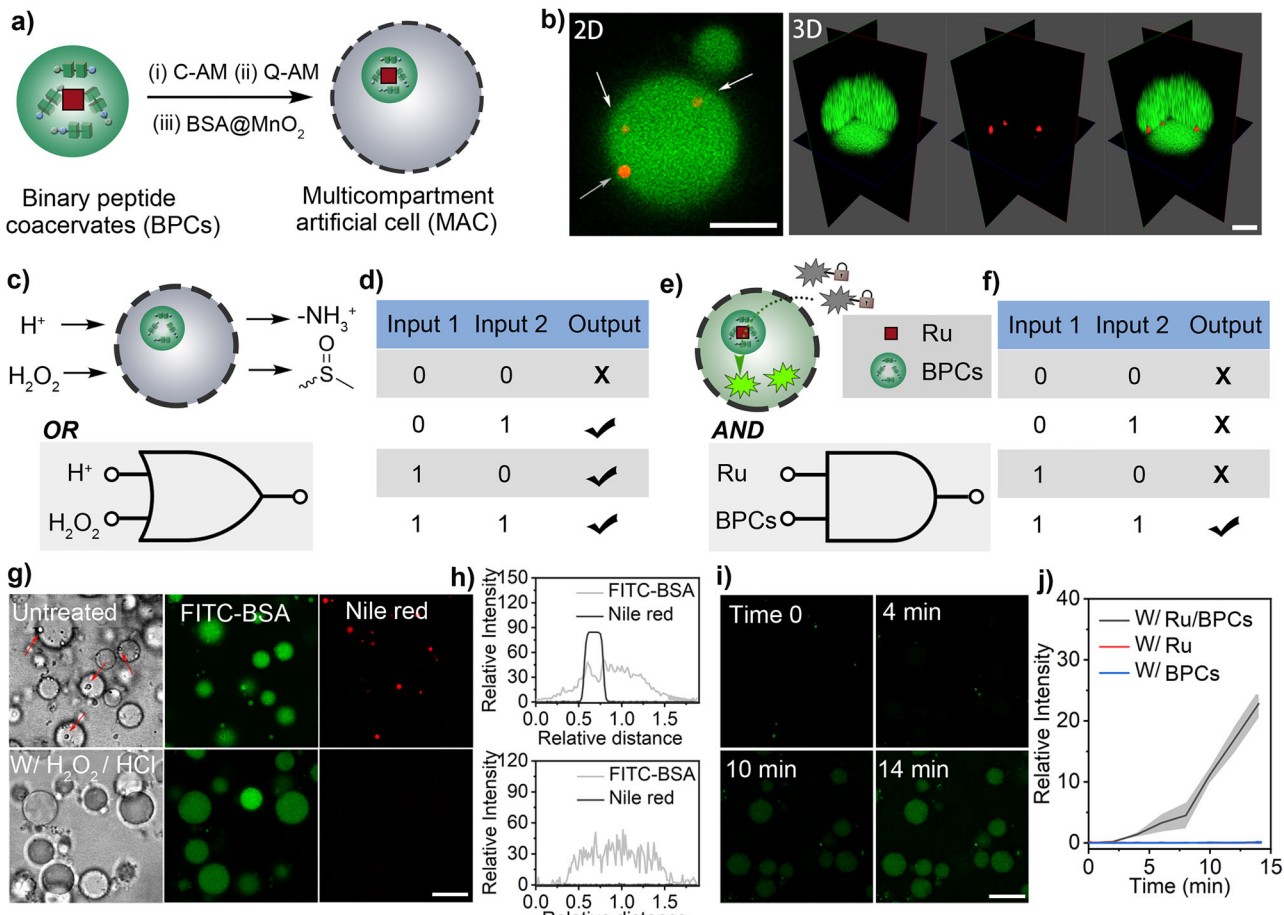

**Fig. 7 | Biomimetic logic-gate engineering with peptide organelles and complex coacervate-based synthetic cells. a** Schematic representation of the formation of multi-compartment artificial cell (MAC). Binary peptide coacervates (FFM/MFF, 1:1 weight ratio) are integrated as internal compartments within membrane-bound complex coacervates. **b** Confocal imaging of MACs. 2D image and 3D reconstruction image showing the BPC inside the complex coacervates, white arrows indicate the BPCs. Scale bar = 5 μm. 3 experiments were repeated independently with similar results. **c** Logic gate function in MACs. Representation of an OR logic gate with two distinct chemical inputs and one output (disassembly of internal BPC). **d** Response to inputs for the OR logic gate. The presence of H⁺ (input 1), H₂O₂ (input 2), or both inputs induces the disassembly of the internal BPC (output). **e** Representation of an AND logic gate using MACs. **f** The AND gate consists of two inputs: the presence of Ru catalyst (input 1), and the presence of BPCs (input 2). The output consists of the occurrence of a Ru-catalysed decaging reaction. Only the presence of both inputs

results in green emissions within the multi-compartment system. **g** Confocal images before and after treatment: Confocal images of the multi-compartmentalized synthetic cells before and after treatment with H₂O₂ and HCl. The signals of Nile Red, which stains peptide organelles, decreased significantly after treatment. 3 experiments were repeated independently with similar results. Scale bar = 10 μm. **h** Plotted relative intensity: Relative fluorescent intensity of FITC-BSA and Nile Red before and after treatment (as shown in **g**). **i** Confocal images showing product formation: The product of the uncaging reaction is generated only in the multi-compartment system where both inputs are present. Three experiments were repeated independently with similar results. Scale bar = 10 μm. **j** Plot of fluorescence intensity of the uncaged product in the multi-compartment system compared with the controls (Ru catalyst only, or BPCs only), data represent mean ± SD for *n* = 5 representative microscopic images. Error bars (grey shaded area) represent the standard deviation (*n* = 5) from confocal imaging analysis.

biomolecular condensates by demonstrating the potential of protein-derived short peptide-based systems to mimic complex cellular behaviour. Binary peptide coacervates exhibited highly tunable dynamics in response to various triggers, including pH, heating, organic solvents, urea and mild oxidation chemistry. These results highlight the versatility and adaptability of peptide-based coacervates in mimicking the dynamic and reversible nature of natural biomolecular condensates. In addition, binary peptide coacervates exhibited sequestration and partitioning capabilities for a wide range of guest molecules, acting as microreactors to promote catalytic activity and synthesis of complex compounds in aqueous media. These peptide-based condensates were further integrated as sub-compartments within a model membrane-bound synthetic cell, resulting in a multi-compartment system of multiphase coacervates that resembles the hierarchical organization within eukaryotic cells. This biomimetic multi-compartment system also allowed us to design Boolean logic

gates (OR and AND) using biochemical inputs. The ability to control assembly and disassembly through various environmental triggers and the integration of functional sub-compartments within synthetic cells opens alternative avenues for understanding intracellular phase separation mechanisms and their role in cellular processes. In addition, this research paves the way for the development of sophisticated biomimetic systems with potential applications in synthetic biology, drug delivery and biochemical device design, thereby extending the practical utility of biomolecular condensates in both scientific research and technological innovation.

## Methods

### Materials

Derivatives of diphenylalanine-based peptides and substrates were synthesized according to literature reports with slight modifications. All solvents and chemicals were used as received.

## Preparation of binary peptide coacervates

Using the coacervate droplet formation from FFM and MFF as an example, FFM and MFF solids were dissolved in 5 mM HEPES buffer (pH ~6) at a concentration of 20 mg/mL after undergoing a heating and cooling process. For microscopy imaging of the coacervates, 1.25 μL of the FFM solution and 1.25 μL of the MFF solution were mixed, followed by the addition of 0.1 M NaOH to adjust the pH above 7, resulting in a final peptide concentration of 5 mg/mL. The solution immediately turned milky, indicating coacervate formation. The presence of binary peptide-based coacervate droplets was confirmed using a Leica DMi8 inverted microscope.

## Turbidity measurement

All turbidity-based titrations were conducted using a Tecan multimode plate reader. Turbidity served as an indicator of phase separation in binary peptides, with droplet formation further confirmed by optical microscopy. Absorbance at 600 nm was used for all turbidity measurements, which were performed at room temperature. After sample addition and shaking for 5 sec, turbidity values were recorded. A well containing the same volume of buffer was used as a blank.

## Partitioning of guest molecules

All experiments were conducted at room temperature unless otherwise stated. Briefly, binary peptide-based coacervate droplets were prepared by adding 0.1 M NaOH to a binary peptide solution in HEPES buffer. Then, 10 μL of the prepared coacervate solution (10 mg/mL) was mixed with 0.2 μL of dye solution (2 mg/mL in DMSO or Milli-Q) by pipetting. The mixture was then deposited onto a glass surface and covered with a cover glass using a home-made setup (Supplementary Fig. S18). Droplets were subsequently imaged using confocal microscopy with the Leica TCS SP5X system.

## Fluorescence recovery after photobleaching (FRAP)

All experiments were conducted at room temperature. Briefly, binary peptide-based coacervate droplets were prepared by adding 0.1 M NaOH to the peptide solution in HEPES buffer. Then, 20 μL of the prepared coacervate dispersion (5 mg/mL) was mixed with 0.5 μL of rhodamine B solution (0.2 mg/mL in 10% DMSO in Milli-Q) by pipetting. The mixture was then deposited onto a glass surface and covered with a coverslip using a custom-built setup to minimize evaporation over prolonged incubation. For FRAP measurements, the fluorescence intensities of the peptide coacervates were photobleached to 40–50% of their initial level, and the subsequent recovery of fluorescence intensity was monitored using the Leica TCS SP5X system.

## Enzymatic metabolism with peptide coacervates

Enzymatic reactions involving binary peptide coacervates were analyzed using confocal imaging. Briefly, 20 μL of binary peptide coacervate solution (5 mg/mL) in HEPES/NaCl buffer (pH ~8) was mixed with 1 μL of HRP solution (final concentration: 0.0025 mg/mL). The mixture was first treated with 0.2 μL of Amplex Red (20 mg/mL). Subsequently, varying concentrations of $H_2O_2$ were added, and the emission intensity at $\lambda_{em} = 580$ nm was recorded using confocal imaging.

## Bio-orthogonal click chemistry with peptide coacervates

To a 200 μL solution of binary peptide coacervates (5 mg/mL), 1 μL of $Cu(PPh_3)_4Br$ (10 mg/mL in DMSO) was added via pipetting. After equilibrating for 2 min, 1 μL of 3-azido-7-hydroxycoumarin (10 mg/mL in DMSO) and 1 μL of phenylacetylene (10 mg/mL in DMSO) were added. The solution was briefly mixed by pipetting. Since the reaction product is hydrophobic, dissolution in an organic solvent was necessary for proper comparison with the control. Therefore, after a designated incubation period, aliquots (20 μL) were collected and mixed with 30 μL of acetonitrile. The change in fluorescence intensity ($\lambda_{em} = 451$ nm) was then measured using a microplate reader.

## Bio-orthogonal uncaging reaction with alloc-protected Rhodamine 110 utilizing peptide coacervates

The bio-orthogonal catalytic potential of binary peptide coacervates was evaluated by assessing the allylcarbamate cleavage of alloc-protected Rhodamine 110 through both microplate reader assays and confocal imaging studies.

**Microplate reader measurements.** To a 100 μL solution of binary peptide coacervates (5 mg/mL), 0.5 μL of [Cp*Ru(cod)Cl] (Cp* = pentamethylcyclopentadienyl, cod = 1,5-cyclooctadiene; abbreviated as Ru, 10 mg/mL in DMSO) was added via pipetting. After equilibration for 2 min, 0.5 μL of caged Rhodamine 110 (10 mg/mL in DMSO) was introduced, and the solution was briefly mixed by pipetting. Changes in fluorescence intensity ($\lambda_{em} = 525$ nm) were monitored using a microplate reader under periodic shaking.

**Confocal imaging measurements.** A 20 μL solution of binary peptide coacervates (10 mg/mL in HEPES/NaCl buffer, pH ~8) was mixed with 0.1 μL of Ru solution (2 mg/mL in DMSO), followed by the addition of 0.1 μL of caged Rhodamine 110 (2 mg/mL in DMSO). The fluorescence emission intensity at $\lambda_{em} = 525$ nm was then recorded using confocal imaging.

## Membrane-bound complex coacervates formation

The formation of membrane-bound complex coacervates was achieved through initial complex coacervation between Q-Am and C-Am, followed by stabilization using BSA@MnO₂ nanoparticles. Q-Am and C-Am were dissolved in PBS buffer at a concentration of 2.5 mg/mL. Coacervation was induced by mixing Q-Am and C-Am solutions in a 1:1 ratio. Subsequently, varying volumes of BSA@MnO₂ nanoparticles were added to the solution via pipetting. The stability of the BSA@MnO₂-stabilized complex coacervates was then assessed using optical microscopy.

## Integration of peptide coacervates as sub-organelle

Peptide coacervates (8 μL, 10 mg/mL) were first formed by adding a small amount of 0.1 M NaOH to the binary peptide solution. Then, 10 μL of C-Am (2.5 mg/mL in PBS, pH ~9) was added and mixed via pipetting for approximately 10 sec. This was followed by the addition of 12.5 μL of Q-Am (2.5 mg/mL in PBS, pH ~9), which was also mixed via pipetting for ~10 sec. Finally, 4 μL of BSA@MnO₂ (~0.7 mg/mL) was introduced, followed by an additional ~10 sec of pipetting to achieve surface stabilization. The resulting solution was then examined using optical microscopy to observe the multi-compartmentalized structure.

## Bio-orthogonal uncaging reaction in the multi-compartmentalized system

The fabrication procedure followed the same steps as described previously, with one key modification: before encapsulation into complex coacervates as organelles, peptide coacervates were pre-treated with 0.2 μL of Ru (2 mg/mL in DMSO). After the formation of the multi-compartmentalized structure, 0.2 μL of caged Rhodamine 110 (2 mg/mL in DMSO) was introduced. The fluorescence emission from the uncaged product was then monitored using confocal microscopy.

## Data availability

All data are available from the corresponding author. Source data are provided with this paper.

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

## Acknowledgements

This work is part of the research conducted within the Max Planck Consortium for Synthetic Biology (MaxSynBio) jointly funded by the Federal Ministry of Education and Research of Germany (BMBF) and the Max Planck Society. S.C. thanks the Alexander von Humboldt Foundation for a fellowship and financial grant (No. 3.5-CHN–1222717-HFST-P), National Natural Science Foundation of China (No. 52403198) and the Fundamental Research Funds for the Central Universities (No. YJ242450). P.Z. and X.Y acknowledge the financial support for this research from the National Natural Science Foundation of China (No. 22372173, 22232006, and 22025207). Allocations of computer time from the Supercomputing Centre and ORISE system in the Computer Network Information Centre at the Chinese Academy of Sciences is gratefully acknowledged.

## Author contributions

S.C., K.L., and L.C.S. designed the research. S.C., P.Z., G.S., and T.I., performed the experiments. S.C., P.Z., X.Y., L.C.S., and K.L. wrote the manuscript. All authors (S.C., P. Z., G.S., T.I., X.Y., K.L., and L.C.S.) reviewed the manuscript.

## Funding

## Competing interests

The authors declare no competing interests.
