## [Transparent Peer Review file · Nature Communications]

Binary Peptide Coacervates as an Active Model for Biomolecular Condensates

Corresponding Author: Dr Lucas Caire da Silva

Version 0:

Reviewer comments:

Reviewer #1

(Remarks to the Author)

This manuscript reports an extensive study of liquid-liquid phase separation (coacervation) in mixtures of diphenylalanine-based peptides. A simplified approach is adopted to probe coacervate droplet formation using light microscopy. A large number of pairs of peptides are screened and there are kinetic studies as a function of pH and thermal treatment as well as methionine oxidation. Following this, selected applications are highlighted, the first being demonstration of the uptake of a wide range of fluorescent dyes (imaged by confocal fluorescence microscopy). Then confined catalysis reactions are demonstrated using a model click reaction and a ruthenium-catalysed deallylation. Finally, multicompartment logic gates in response to different inputs (pH shift, oxidation etc).

This manuscript builds on their recent Nature Comm. Article 10.1038/s41467-023-44278-9 which discusses diphenylalanine-base coacervates and introduces many of the application tests employed in the new submission, i.e. dye uptake, catalysis test as well as multicompartment cells. There is thus some similarity in the manuscripts although the new one describes distinct systems (mixtures) and there is the novel aspect of the logic gate, as well as some different examples of catalytic reactions (click reaction). The discussion and analysis are generally clear and the experimental results are convincing, however I found the simulations less valuable due to the somewhat unrealistic constraints imposed (small number of molecules, periodic constraints) as well as unclear definitions, fibrils versus droplets.

The manuscript is quite long and there are a huge number of Supporting Figures which does not enhance readability. In regard to the latter, it does serve to overwhelm the reader/reviewer with evidence to show the exhaustive range of work undertaken and it is not easy to suggest specific cuts to the number of Supporting Figures without omitting relevant data. This is overall a very strong study of interesting systems with remarkable applications that will stimulate further interest in the already highly active field of model biomolecular condensate systems. The potential future applications in biocatalysis, in particular, are very exciting.

I recommend publication in Nature Comms, subject to addressing comments and making revisions as detailed below.

1. On p.6, the SI Figures should be cited in order and all should be cited.
2. On p.7, does the mixing method influence the observations, i.e. does the order of mixing affect the coacervation?
3. Is there any information on the peptide distributions in the droplets compared to the external phase? On p.9 there is an NMR estimation of overall contents of peptides in mixtures but nothing on the segregation in different phases.
4. In Fig.2 caption, the scale bars are omitted in part (b).
5. It would be good to have example phase diagrams showing conditions for coacervation as a function of concentration and pH
6. Figure 3, please clarify how 'fibril' and 'condensate' are defined. In part (c) the yellow region cannot be seen clearly and in part (d) there is no definition of the colour scale (z-axis), it would also be better if the x- and y-axes were over the same range.
7. Line 353: It mentions "reversible oxidation", but reversibility is not shown in Fig.4d.
8. There is a lot of self citation and this should be balanced with references to other groups e.g. on catalysis and compartmentalized reactors.
9. It is hard to see features in the confocal images for Nile red (N.B spelling of dye name in text in several places) (Fig.7g). Several errors in the caption of Fig.7 should be corrected: "indicate", "Nile red fluorescence intensity", add label for part j).
10. It should be "Conclusions" section.

11. In the SI experimental section, details of FRAP experiment should be added.
12. In Fig.S44, the units for urea concentration differ from those used for hexanediol in Fig.S43.
13. Fig.S43 and Fig.3a should have scale bars. The constraint of periodicity does not look realistic for coacervate droplet formation, nor fibril placement.
14. The contour levels in Fig.S45 should be defined.
15. Throughout, there are a number of typographical errors that should be corrected, along with errors in reference author names etc that should be carefully checked.

Reviewer #2

(Remarks to the Author)

In this paper, Cao et al. reported the formation of coacervate droplets through the liquid-liquid phase separation of binary short peptides. The peptide coacervates exhibited dynamic, liquid-like characteristics, as demonstrated by fluorescence recovery after photobleaching (FRAP) and droplet fusion studies. The coacervates were shown to enrich hydrophobic cargos and function as microreactors, enhancing catalytic activity and facilitating the synthesis of complex compounds in aqueous media. Additionally, the coacervates displayed stimuli responsiveness in the presence of various triggers, including pH, heat, organic solvents, NaCl, 1,6-hexanediol, urea, and hydrogen peroxide. The authors also utilized these peptide coacervate droplets to construct multi-compartment systems that mimic cellular structures. I believe this study advances our understanding of phase separation behavior among multi-component biomolecules.

- 1) The use of multicomponent peptides to prepare coacervate droplets is novel to me. However, the experiments involving guest molecule sequestration, droplet fusion, confined chemical reactions, stimuli-responsive properties, Boolean logic gates, and multi-compartments are of less novelty and predictable. These properties are typically observed in standard polyelectrolyte coacervates. I would expect more information about the peptide structure-property relationships.
- 2) The authors demonstrated the coacervate formation of FFM/MFF. Will the coacervate microdroplets formed with FFM/FMF or MFF/FMF?
- 3) I would expect the authors to provide guidelines for designing multicomponent coacervates. How does the amino acid sequence affect coacervation? For example, if methionine is substituted with another amino acid, what changes might occur?
- 4) Analyzing the composition of multi-component coacervates is crucial. The authors indicated that the FFM/MFF molar ratio within the coacervates closely resembles the 1:1 feed ratio. What would the peptide ratio be if the feed ratio changes, such as to 40% and 60% FFM?
- 5) Following the previous question, what is the FFM/FL ratio inside the coacervates? Is it close to 1:1?
- 6) In all the combinations for coacervates, the peptide contained hydrophobic amino acids such as phenylalanine and leucine. Will coacervates be formed if one peptide is hydrophobic and the other is hydrophilic, such as MFF/MGG or MFF/MFG?
- 7) Is there experimental evidence of interactions between FFM and LF?
- 8) The setup involving the cover slip is unclear. In the Supporting Information, it states, "Then the mixture was dropped on a glass surface with a cover glass using a homemade setup (Figure S16)." More details about this setup are required.
- 9) The microscopic images at pH 6 in Figures 27, 28, and 29 appear to be duplicated. Please double-check this.
- 10) The nomenclature for the short peptides is confusing. For example, F represents phenylalanine in FFG, while G does not correspond to glycine. Similarly, the letter "I" in FFI does not stand for isoleucine.

Reviewer #3

(Remarks to the Author)

This work reports the design of peptide-based simple coacervates using mixtures of short peptides containing di-phenylalanine motifs. The authors show that single-peptide mixture rapidly evolve into fibrous aggregates when exposed to phase-separating conditions (mainly by changing the pH of the solution), while mixtures of 2 or more peptides phase separate into liquid-like droplets that persist in a liquid state for extended periods. The key hypothesis for the absence of fiber formation is linked to the disruption of homo-peptide interactions, which is supported by numerical simulations. Similarly to their recently reported study, the authors illustrate that the hydrophobic environment of these peptide-based coacervates favors chemical reactions. They last extend the use of these systems to produce hierarchical coacervates able to respond to external stimuli as Boolean OR and AND logic gates.

This work builds upon an emerging body of literature focusing on the design of simple coacervates based on minimal peptide sequences. It adds valuable insights to previous research, including by the same authors, specifically showing that peptide mixtures alleviate aggregation issues in these simple coacervates. However, I believe a few points should be revised to strengthen the authors' conclusions.

i/ Stabilization of the liquid-like phase with peptide mixtures

- While I agree that coacervates remain spherical at microscopic length-scales without observable fibers even after a few days, I suggest a more detailed characterization of the liquid-like behavior of the droplets over time. It is possible that coacervate properties evolve due to internal restructuring at sub-micron scales, which might not be observable through microscopy. For example, does molecular mobility assessed by FRAP or fusion dynamics change over time? Repeating these experiments at different time points after the formation of the coacervates (up to 3 days) could provide clarity.
- Related to this point, no values for the viscosity-to-surface tension ratio are provided from fusion experiments. It would be

informative to compare these values with other coacervates and condensates.

- Also, line 178, the author mention the “formation of relatively stable coacervate droplets”. What is meant by “relatively”?
- The FRAP experiments are performed using a low molecular weight dye (rhodamine B). The recovery appears slow for such a small molecule, suggesting low mobility within the droplets. Could the authors please comment on this?

ii/ Clarification of the role of electrostatics. The authors mention that “the electrostatic repulsion between the peptides is screened by the electrolyte, favouring phase separation” (lines 241-242), but also explain that phase separation is promoted at pH 9 due to charge neutralization of the amine groups (lines 164 and following). This discussion could benefit from clarification. Providing the pKa of the amine groups in the different peptides used would help resolve this.

iii/ Temperature responsiveness. Could the authors comment on whether the observed behavior depends on the heating rate? Additionally, is there any hysteresis when cooling down the solution?

iv/ Partitioning of dyes, enzymes and enzyme reactions.

- The graph shown in Figure 5g is potentially misleading, as it gives the impression of a trend in the partition coefficients. How are the cargoes classified along the x-axis? I suggest avoiding ranking them by decreasing partition coefficients unless this reflects an intrinsic property of the dyes (e.g., hydrophobicity or charge). Instead, classifying them as neutral, positive, and negative might be clearer.

- Cy5-GOx appears to accumulate at the droplets interface. Could the authors comment on this observation?

- Lines 410-411 state that “Confocal imaging of the reaction revealed the production of fluorescent resorufin ($\lambda_{em} \sim 580$ nm) confined within the coacervates”. However, there is no direct evidence that resorufin is only produced within the droplets. It is possible that the reaction occurs outside, with the product subsequently diffusing and accumulating in the droplets. Could the authors address this point?

v/ Comparison with “conventional” coacervates (lines 441-455). The term “conventional” coacervates (lines 441-455) seems ambiguous, and I am not sure whether the coacervates used here as a single example reflect broader trends in complex coacervates. Complex coacervates are known to sequester both hydrophobic dyes and hydrophilic cargoes, and these sequestration properties can be highly specific and system-dependent. For this reason, I would recommend removing this direct comparison between peptide coacervates and the specific amylose coacervates. If necessary, this discussion could be reserved for later when addressing hierarchical coacervate assembly.

Version 1:

Reviewer comments:

Reviewer #1

(Remarks to the Author)

The authors have carefully revised their manuscript and addressed my comments. I therefore recommend acceptance without further revision

Reviewer #2

(Remarks to the Author)

The authors have satisfactorily addressed my main concerns, and the paper is now ready for publication.

Reviewer #3

(Remarks to the Author)

The authors have addressed my comments in this revised version. I now support publication in Nature Communications.

REVIEWER COMMENTS

Reviewer #1 (Remarks to the Author):

This manuscript reports an extensive study of liquid-liquid phase separation (coacervation) in mixtures of diphenylalanine-based peptides. A simplified approach is adopted to probe coacervate droplet formation using light microscopy. A large number of pairs of peptides are screened and there are kinetic studies as a function of pH and thermal treatment as well as methionine oxidation. Following this, selected applications are highlighted, the first being demonstration of the uptake of a wide range of fluorescent dyes (imaged by confocal fluorescence microscopy). Then confined catalysis reactions are demonstrated using a model click reaction and a ruthenium-catalysed deallylation. Finally, multicompartment logic gates in response to different inputs (pH shift, oxidation etc).

This manuscript builds on their recent Nature Comm. Article 10.1038/s41467-023-44278-9 which discusses diphenylalanine-base coacervates and introduces many of the application tests employed in the new submission, i.e. dye uptake, catalysis test as well as multicompartment cells. There is thus some similarity in the manuscripts although the new one describes distinct systems (mixtures) and there is the novel aspect of the logic gate, as well as some different examples of catalytic reactions (click reaction). The discussion and analysis are generally clear and the experimental results are convincing, however I found the simulations less valuable due to the somewhat unrealistic constraints imposed (small number of molecules, periodic constraints) as well as unclear definitions, fibrils versus droplets.

Response:

We thank the reviewers for their insightful comments. We acknowledge the significant gap between full-atom MD simulations and experimental phenomena, both in terms of time and spatial scales. Given that both single-peptide and binary-peptide systems can form condensates at early stages and transition to nanofiber formation after prolonged incubation, it is challenging to distinguish between condensate and fiber formation based solely on snapshots from MD simulations. Therefore, the analysis of aggregation propensity, clustering degree, intermolecular interactions, and hydrogen bonding is crucial for understanding the molecular mechanisms underlying the differences in phase behavior. In this revision, we have removed unrealistic representations of periodic simulation boxes and provided a clearer definition of condensates and fibers based on aggregation propensity and clustering degree values.

The manuscript is quite long and there are a huge number of Supporting Figures which does not enhance readability. In regard to the latter, it does serve to overwhelm the reader/reviewer with evidence to show the exhaustive range of work undertaken and it is not easy to suggest specific cuts to the number of Supporting Figures without omitting relevant data.

This is overall a very strong study of interesting systems with remarkable applications that will stimulate further interest in the already highly active field of model biomolecular condensate systems. The potential future applications in biocatalysis, in particular, are very exciting.

I recommend publication in Nature Comms, subject to addressing comments and making revisions as detailed below.

Response:

We thank the reviewer for the helpful and constructive comments. We have carefully revised the manuscript to address the reviewer's comments.

1. On p.6, the SI Figures should be cited in order and all should be cited.

Response:

The manuscript has been revised and updated accordingly.

2. On p.7, does the mixing method influence the observations, i.e. does the order of mixing affect the coacervation?

Response:

Peptide mixtures are typically prepared at acidic pH, allowing the peptides to dissolve and form a homogeneous solution. Subsequently, droplets of NaOH solution are added to the peptide solution via pipetting to induce peptide desolvation and subsequent coacervation. Therefore, the order of mixing is not expected to have a significant impact on the coacervation behavior.

3. Is there any information on the peptide distributions in the droplets compared to the external phase? On p.9 there is an NMR estimation of overall contents of peptides in mixtures but nothing on the segregation in different phases.

Response:

To answer this question, the peptide distributions in the coacervate droplets and the surrounding diluted solution were determined by HNMR measurements after low-speed centrifugation and re-dispersing. 1,3,5-Trimethoxybenzene was added to the HNMR samples as an external standard. According to the proton integrations (aromatic units), the peptide contents in the coacervate phase and the surrounding diluted phase are calculated with a weight ratio of ~1.25:1 (dense:dilute). This data has been included in Supplementary Figure 41.

The following text has been added to the revised manuscript:

“The peptide content in the concentrated coacervate phase and the surrounding dilute phase was estimated using NMR with 1,3,5-trimethoxybenzene as an external standard. The results indicated that the peptide content ratio between the concentrated and diluted phases was approximately 1.25:1 (Supplementary Figure 41). However, due to the substantial difference in volume between bulk phase and the coacervate phase, the peptide concentration in the coacervate would be much higher than that in the surrounding diluted phase. Additionally, when mixtures of FFM and MFF were prepared with initial FFM proportions of 40% or 60% prior to coacervation, the calculated FFM ratios in the coacervate phase were ~ 40.5% and 58.3%, respectively, closely matching the initial feed ratios (Supplementary Figure 42). Based on these results, a 1:1 weight ratio of MFF to FFM was used for the subsequent studies.”

4. In Fig.2 caption, the scale bars are omitted in part (b).

Response:

Scale bar in Fig. 2 has been changed from black to white for improved visibility.

5. It would be good to have example phase diagrams showing conditions for coacervation as a function of concentration and pH

Response:

Phase diagrams of peptide coacervation has been added in the revised supporting information (Supplementary Figure 45).

6. Figure 3, please clarify how ‘fibril’ and ‘condensate’ are defined. In part (c) the yellow region cannot be seen clearly and in part (d) there is no definition of the colour scale (z-axis), it would also be better if the x- and y-axes were over the same range.

Response:

We thank the reviewers for the insightful question.

(1) We have included a clearer definition of condensates and fibers based on the aggregation propensity (AP) and clustering degree (CD) values, based on our previous studies [Matter 6, 1945-1963, 2023] and those of Prof. Guanghong Wei's group's [Cell Rep. Phys. Sci. 2, 100579 (2021)].

The following text has been added to the revised manuscript:

“Based on these parameters, it can be concluded that the FFM+MFF system exhibits a lower degree of clustering and reduced aggregation propensity compared to pure MFF or pure FFM systems. This is consistent with a relatively poor fiber-forming ability. (Figure 3b / Supplementary Figure 53). Based on the coarse-grained MD simulations of dipeptides performed by Tang et al, the fiber formation is characterized by $AP > 3$ and $CD > 0.9$, while condensate formation is defined by $1 < AP < 3$ and $CD > 0.5$.⁶⁰ However, in our previous all-atom MD simulations of z-FF peptides, the boundary between condensates and fibers shifted to lower AP values. In these simulations, condensates were stable when the SASA ranged from 0.65 to 0.85 (equivalent to $1.2 < AP < 1.6$), and fiber formation occurred when the SASA dropped below 0.65 (equivalent to $AP > 1.6$).⁵⁰ Therefore, in this study, we propose revised definitions: fibers are defined as $AP > 1.6$ and $CD > 0.9$, while condensates are defined as $1.2 < AP < 1.6$ and $CD > 0.5$.”

(2) Figure 3c has been redrawn and the yellow region can be seen clearly now.

(3) The color scales (z-axis) in Figure 3 have been added.

(4) The x- and y-axes in Figure 3 were set with the same range.

7. Line 353: It mentions “reversible oxidation”, but reversibility is not shown in Fig.4d.

Response:

To avoid confusion, the sentence was revised as follows:

“Thus, the responsive behavior of thioether groups in FFM/MFF was investigated to actively control their coacervation, similar to post-translational modifications used to regulate protein coacervation in cells (Figure 4d).²³”

8. There is a lot of self citation and this should be balanced with references to other groups e.g. on catalysis and compartmentalized reactors.

Response:

To address the comment, more citations on catalysis and compartmentalized reactors have been added and updated in the revised manuscript.

9. It is hard to see features in the confocal images for Nile red (N.B spelling of dye name in text in several places) (Fig.7g). Several errors in the caption of Fig.7 should be corrected: “indicate”, “Nile red fluorescence intensity”, add label for part j).

Response:

The contrast of Nile red emission in Figure 7g has been enhanced, and the caption of Figure 7 has been revised in the updated manuscript.

10. It should be “Conclusions” section.

Response:

This has been updated in the revised manuscript.

11. In the SI experimental section, details of FRAP experiment should be added.

Response:

The details of FRAP experiments have been updated in the Supplementary Information.

“Fluorescence recovery after photobleaching (FRAP): All experiments were performed at room temperature. Briefly, the coacervate droplet solution was first prepared by adding 0.1 M NaOH solution to the peptide solution in HEPES buffer. Then, 20 μL of the prepared coacervate dispersion (5 mg mL^{-1}) was mixed with 0.5 μL rhodamine B solution (0.2 mg mL^{-1} in 10% DMSO in Milli-Q) by pipetting. The mixture was then dropped onto a glass surface with a coverslip using a homemade setup to avoid evaporation after days of incubation. For FRAP measurements, the fluorescence intensities of the peptide coacervates were quenched to 40%-50% of the initial level, and then the recovery of the fluorescence intensity of the droplets was followed using the Leica TCS 264 SP5X system.”

12. In Fig.S44, the units for urea concentration differ from those used for hexanediol in Fig.S43.

Response:

The units for urea and hexanediol concentration have been standardized to be consistent (i.e., mM).

13. Fig.S43 and Fig.3a should have scale bars. The constraint of periodicity does not look realistic for coacervate droplet formation, nor fibril placement.

Response:

Scale bars have been added, and the periodic representation of simulation of simulation boxes has been removed.

14. The contour levels in Fig.S45 should be defined.

Response:

The color scale has been added in the revised version.

15. Throughout, there are a number of typographical errors that should be corrected, along with errors in reference author names etc that should be carefully checked.

Response:

The revised manuscript has been thoroughly revised. We hope that the revised version will be suitable for publication.

Reviewer #2 (Remarks to the Author):

In this paper, Cao et al. reported the formation of coacervate droplets through the liquid-liquid phase separation of binary short peptides. The peptide coacervates exhibited dynamic, liquid-like characteristics, as demonstrated by fluorescence recovery after photobleaching (FRAP) and droplet fusion studies. The coacervates were shown to enrich hydrophobic cargos and function as microreactors, enhancing catalytic activity and facilitating the synthesis of complex compounds in aqueous media. Additionally, the coacervates displayed stimuli responsiveness in the presence of various triggers, including pH, heat, organic solvents, NaCl, 1,6-hexanediol, urea, and hydrogen peroxide. The authors also utilized these peptide coacervate droplets to construct multi-compartment systems that mimic cellular structures. I believe this study advances our understanding of phase separation behavior among multi-component biomolecules.

Response:

We sincerely appreciate the constructive feedback and insightful comments provided by the reviewer. The manuscript has been carefully revised to address all the concerns raised.

1) The use of multicomponent peptides to prepare coacervate droplets is novel to me. However, the experiments involving guest molecule sequestration, droplet fusion, confined chemical reactions, stimuli-responsive properties, Boolean logic gates, and multi-compartments are of less novelty and predictable. These properties are typically observed in standard polyelectrolyte coacervates. I would expect more information about the peptide structure-property relationships.

Response:

We appreciate the reviewer's thoughtful comments and their insights into the experiments involving guest molecule sequestration, droplet fusion, confined chemical reactions, stimuli-responsive properties, Boolean logic gates, and multi-compartments. We understand the reviewer's observation that these properties are often found in polyelectrolyte coacervates. While we acknowledge that these behaviors—such as guest molecule sequestration, fusion, and stimuli-responsive properties—are typically observed in polyelectrolyte coacervates, we would like to emphasize that the use of multicomponent peptides to prepare coacervate droplets introduces several unique features. Peptides offer precise control over sequence and chemical functionality, enabling more tunable structure-property relationships that cannot be easily replicated by traditional polyelectrolytes. The modularity and molecular diversity inherent to peptides allow us to finely adjust properties such as charge distribution, hydrophobicity, and secondary structure, resulting in coacervates with tailored functionality that go beyond standard polyelectrolyte systems. These unique features make peptide-based coacervates draw considerable interest from the biomimicry community, with an increasing number of published works not only in the field of synthetic biology, but also in the fields of catalysis and biomedical engineering.

The novelty of our work lies in the rational design of multicomponent peptides that lead to coacervation, combined with the ability to manipulate peptide sequence to control phase behavior, responsiveness, and molecular interactions. Conventional peptide-based coacervates were usually fabricated from polypeptides with complicated molecular composition and very large molecular size (Nat. Commun. 2020, 11, 5949 / J. Am. Chem. Soc. 2021, 143, 18196 / Chem. Soc. Rev. 2021,50, 3690). Engineering coacervate droplets from short peptide species with simplified structures and small molecular size have been reported only in a very recent cases (Nature Chemistry, 2021, 13, 1046 / J. Am. Chem. Soc. 2022, 144, 15155 / Adv. Mater. 2022, 2202913). In addition, the peptide coacervation behaviors were usually limited to specific composition and sequence design, which make it challenge to precisely manipulate the properties of the coacervate droplets.

In this manuscript, we showed a more robust strategy for the formation of short-peptide based coacervate droplets, which allowed for dynamic fine-tuning of properties like tailored responsiveness (dynamic assembly and morphology transition), as well as specific non-covalent interactions (such as hydrogen bonding, hydrophobic interactions, and π - π stacking). The concept and design of short peptide-based coacervates are supposed to provide insight in the motifs and interactions that drive liquid-liquid phase-separation in intrinsically disordered proteins/regions, which is important for understanding the role of biomolecular condensates in cellular functions. In addition, dynamic and subtle control of short peptides, even at the single amino acid level, directly dictate the supramolecular structure and material properties, allowing the

establishment of sequence-structure and structure-function relationships (Molecular Cell 2022, 82, 3193–3208 / Nat. Commun. 2023, 14, 421). These coacervates present unprecedented versatility in biological mimicry and potential applications for bio-inspired systems

We have now added additional discussion in the manuscript that highlights the connection between specific peptide sequences and the resulting coacervate properties. For example, we describe how variations in peptide hydrophobicity, sequence and structure propensity influence the phase-separating behaviors of peptides and the formation of peptide droplets. We hope that these clarifications address the reviewer's concerns and provide a clearer understanding of the novelty and significance of our work.

2) The authors demonstrated the coacervate formation of FFM/MFF. Will the coacervate microdroplets formed with FFM/FMF or MFF/FMF?

Response:

In response to the comments, FMF peptide was synthesized and the coacervation behavior was investigated through mixing with FFM or MFF. Similar to MFF and FFM, the FMF peptide was also soluble at acidic conditions (~pH 6). Interestingly, FMF peptide alone formed relatively stable coacervate droplets, with no significant liquid-to-solid transition observed. When mixing with FFM or MFF, stable coacervate droplets were also observed, supporting the notion that the prevention of fiber formation due to compositional complexity is general characteristic of coacervates involving short peptide sequences. The results from this additional experiment are presented in the as newly added Supplementary Figure 35.

The following text has been added to the manuscript:

“To provide additional evidence, we mixed a phase-separating peptide, FFM or MFF, with a coacervate-forming peptide, FMF. FMF is soluble at pH 6 and forms relatively stable coacervate droplets at pH 9. When mixed with FFM or MFF, the mixtures also produced spherical droplets that did not transform into irregular aggregates or fibrous structures during observation (Supplementary Figure 35). This result indicates that the stability of short-peptide coacervates is largely dependent on the compositional complexity of the coacervate phase. In the case of FFM, the complexity is achieved by simply rearranging the amino acids within the short peptide sequences. However, the specific identity of the components also plays an important role, as demonstrated by mixing the phase-separating peptide FFM with the non-phase-separating peptide LF.”

3) I would expect the authors to provide guidelines for designing multicomponent coacervates. How does the amino acid sequence affect coacervation? For example, if methionine is substituted with another amino acid, what changes might occur?

Response:

While a single peptide (FFM, MFF, FFI, FFG, and FFF in the initial submission) form irregular aggregates or fibrous structure at basic pH conditions, a completely different behaviour was observed when binary peptide mixtures were investigated. We found that coacervate droplets formed without transforming into fibres or solid aggregates when two or more short peptides were present in the solution. For example, a 1:1 mixture (weight ratio) of FFM and MFF, which differ only in the position of the methionine amino acid, formed a transparent solution at pH 6. When the pH was increased to approximately 9, microscopy images revealed the formation of relatively stable coacervate droplets without conversion to fibres or solid aggregates (Supplementary Figure 39). The behaviour of the MFF/FFM mixture differs from that of solutions containing only MFF or FFM, which form unstable coacervates that rapidly transform into fibres and solid aggregates (Supplementary Figure 17 / 19).

To determine whether the higher stability of coacervates from binary peptide mixtures is a general property, we investigated different combinations and tested their phase separation behaviour. We observed the formation of stable liquid coacervates (lasting over 30 minutes) in each two-peptide 1:1 mixture (weight ratio) containing FFM, FFIA, FFF, FFFE, and MFF (in the updated Supplementary Figure 23 to 31). Coacervates

with improved phase stability were also observed when mixing up to five peptides, allowing the formation of peptide coacervates with more complex compositions (Supplementary Figure 32, 33 and 34). The formation of liquid coacervates in mixtures of phase-separating peptides may be due to the disordered packing of peptide molecules in the condensed state, which is likely to inhibit their tendency to form solid aggregates such as fibres (as suggested by the MD simulations).

To show more evidence of the peptide sequence effect in the coacervation, several new peptides including FMF, MGG and MFG have been synthesized and their coacervation behaviors in the peptide mixtures have been investigated and updated in the revised manuscript (Supplementary Figure 35 / 37 / 38).

4) Analyzing the composition of multi-component coacervates is crucial. The authors indicated that the FFM/MFF molar ratio within the coacervates closely resembles the 1:1 feed ratio. What would the peptide ratio be if the feed ratio changes, such as to 40% and 60% FFM?

Response:

The effect of varying the weight ratio of FFM and MFF (from 9:1 to 1:9) was investigated in the initial submission (now updated as Supplementary Figure 22). Their phase separation behaviors were recorded using microscopic imaging. The results showed that the FFM ratio had a significant effect on the resulting phase properties. For example, solutions with FFM weight ratios between 30% and 70% produced relatively stable liquid coacervates (Figure 2a / Supplementary Figure 22). In contrast, when the FFM weight ratio was below 30% or above 70%, i.e., when there was an excess of any of the two peptides, the initially formed binary peptide coacervates gradually transformed into either irregular solid aggregates or fibrous structures within 20 minutes of observation (Figure 2a / Supplementary Figure 22).

Thus, we found the peptide mixtures with a ratio of 40% or 60% FFM also formed relatively stable coacervate droplets which did not transform into fibrous structure or irregular aggregates after prolonged incubation.

To further provide additional information in the cases of 40% and 60% FFM, HNMR measurements were utilized to determine the peptide contents. As displayed in the newly added Supplementary Figure 42, when 40% FFM was fed in the mixture of FFM/MFF coacervates, the content of peptides in the coacervate phase was determined to be 40.5% FFM and 59.5% MFF. In addition, when 60% FFM was fed in the mixture of FFM/MFF coacervates, the content of peptides in the coacervate phase was determined to be 58.3% FFM and 41.7% MFF, in the newly added Supplementary Figure 42.

5) Following the previous question, what is the FFM/FL ratio inside the coacervates? Is it close to 1:1?

Response:

The solution of pure FFM upon the increase of pH initially formed small droplets, which very fast transformed into fiber like structure (Supplementary Figure 17). And FL, which should be LF actually in the manuscript, is soluble at both pH 6 and pH 9, with no visible droplets or aggregates observed in the microscopic images (shown in Supplementary Figure 36). When FFM and LF were mixed, coacervates initially formed when the pH was raised to around 9, but these droplets quickly (<5 min) transformed into a fibrous structure, very similar to the results obtained from single peptide FFM. The transient nature of the coacervate droplets obtained with FFM/LF prevent the determination of the FFM/LF content of the metastable and short-lived coacervate droplets.

The phase separation behavior in the mixture of FFM and LF is different from the coacervation behavior when mixing two phase separating peptides (e.g., FFM, MFF, FFIA, FFF, FFE). Based on this experimental evidence, it has been demonstrated that the mixture of FFM/LF does not result in coacervation droplets after incubation for >5 min but forms a fibrous structure very similar to that of FFM alone. The formation of the fibrous structure is associated with intermolecular aromatic and hydrogen bonding interactions between FFM peptides, which drive the alignment of peptide molecules and ordered packing (Angew. Chem. Int. Ed. 2019, 58, 18116). The resulting fiber structure indicated a very low distribution level of LF molecules in the initial FFM coacervates, which did not affect the subsequent fiber transition.

Based on the above discussions, it is believed that LF is largely dispersed in solution rather than partitioned into the initial coacervates or the subsequent fibrous structure.

6) In all the combinations for coacervates, the peptide contained hydrophobic amino acids such as phenylalanine and leucine. Will coacervates be formed if one peptide is hydrophobic and the other is hydrophilic, such as MFF/MGG or MFF/MFG?

Response:

We thank the reviewer for the insightful comments. In response to the issues raised, we newly synthesized MGG and MFG peptides and investigated their coacervation behaviors upon mixing with MFF. Due to their reduced hydrophobicity, neither MFG nor MGG alone formed coacervates under basic conditions. However, when mixed with MFF, the phase-separation behavior was similar to that of MFF alone, initially forming droplets that rapidly transformed into irregular aggregates. These findings have been incorporated into the manuscript as the newly added Supplementary Figures 37 and 38.

7) Is there experimental evidence of interactions between FFM and LF?

Response:

We do not have direct experimental evidence for the interactions. However, our observations indicate that mixing FFM with LF resulted in transient, short-lived coacervation followed by fiber transition, closely resembling the behavior of FFM alone. Given that LF is highly soluble at both pH 6 and pH 9, with no visible droplets or aggregates observed in microscopic images, we hypothesize that LF is too hydrophilic to efficiently partition into FFM droplets and prevent fiber formation.

In contrast, mixing FFM with other phase-separating peptides (e.g., MFF, FFIA, FFF, FFFE), which can form droplets due to their lower solubility, maintained the coacervate state. This suggests that peptide mixtures significantly influence intra- and inter-molecular hydrogen bonding, which is regarded as the primary driving force behind fiber transition (*Angew. Chem. Int. Ed.* 2019, 58, 18116). Therefore, LF is likely to remain largely solubilized in the solution, rather than partitioning into the initial coacervates or subsequent fibrous structures.

8) The setup involving the cover slip is unclear. In the Supporting Information, it states, "Then the mixture was dropped on a glass surface with a cover glass using a homemade setup (Figure S16)." More details about this setup are required.

Response:

Additional details regarding the setup used for droplet imaging have been included and updated in the revised Supplementary Figure 16.

9) The microscopic images at pH 6 in Figures 27, 28, and 29 appear to be duplicated. Please double-check this.

Response:

Under pH 6 conditions, all peptides and their mixtures were completely soluble in the buffer, resulting in very similar images. In addition, defects and dirt on the objective lens caused artifacts in the image backgrounds, such as round or linear features and dots, which further contributed to the similarity of the pH 6 images. To address the reviewer's concerns, all images taken under pH 6 conditions were carefully re-examined. In addition, the images included in the Supplementary Information (SI) have been carefully cropped to eliminate these similar background features. Importantly, this cropping does not alter the results or interpretation of the data, ensuring that the integrity and validity of the results remain unaffected.

10) The nomenclature for the short peptides is confusing. For example, F represents phenylalanine in FFG, while G does not correspond to glycine. Similarly, the letter "I" in FFI does not stand for isoleucine.

Response:

To clarify and avoid confusion, the nomenclature for short peptides has been updated in the revised manuscript. Specifically, "FFG" has been changed to "FFFE-OMe," where "E-OMe" represents glutamic acid with a methyl ester. Similarly, "FFI" has been updated to "FFIba," where "Iba" refers to an isobutylamine tag.

Reviewer #3 (Remarks to the Author):

This work reports the design of peptide-based simple coacervates using mixtures of short peptides containing di-phenylalanine motifs. The authors show that single-peptide mixture rapidly evolve into fibrous aggregates when exposed to phase-separating conditions (mainly by changing the pH of the solution), while mixtures of 2 or more peptides phase separate into liquid-like droplets that persist in a liquid state for extended periods. The key hypothesis for the absence of fiber formation is linked to the disruption of homo-peptide interactions, which is supported by numerical simulations. Similarly to their recently reported study, the authors illustrate that the hydrophobic environment of these peptide-based coacervates favors chemical reactions. They last extend the use of these systems to produce hierarchical coacervates able to respond to external stimuli as Boolean OR and AND logic gates.

This work builds upon an emerging body of literature focusing on the design of simple coacervates based on minimal peptide sequences. It adds valuable insights to previous research, including by the same authors, specifically showing that peptide mixtures alleviate aggregation issues in these simple coacervates. However, I believe a few points should be revised to strengthen the authors' conclusions.

Response:

We are grateful for the constructive comments, and we have made every effort to address them in a systematic fashion.

i/ Stabilization of the liquid-like phase with peptide mixtures

- While I agree that coacervates remain spherical at microscopic length-scales without observable fibers even after a few days, I suggest a more detailed characterization of the liquid-like behavior of the droplets over time. It is possible that coacervate properties evolve due to internal restructuring at sub-micron scales, which might not be observable through microscopy. For example, does molecular mobility assessed by FRAP or fusion dynamics change over time? Repeating these experiments at different time points after the formation of the coacervates (up to 3 days) could provide clarity.

Response:

To address the reviewer's concern, we have included FRAP tests conducted on days 0, 1, and 3 following the formation of coacervates. These updates have been incorporated into the revised manuscript. The plotted profiles from the FRAP experiments demonstrate that the fluorescent recovery behaviors of rhodamine B within the coacervates remain consistent across days 0, 1, and 3, indicating no significant changes in molecular mobility or liquid-like properties during the 3-day incubation period. These findings have been added as Supplementary Figure 49.

The following discussion has been included in the manuscript:

"Furthermore, even after incubation for up to 3 days, the droplets exhibited a level of fluorescence recovery comparable to that of fresh droplets, indicating no significant changes in molecular mobility or liquid-like properties over extended incubation periods (Supplementary Figure 49)."

- Related to this point, no values for the viscosity-to-surface tension ratio are provided from fusion experiments. It would be informative to compare these values with other coacervates and condensates.

Response:

We thank the reviewer for the insightful comments. The viscosity-to-surface tension ratio (η/γ) has been updated in the revised manuscript and is now presented in Supplementary Figure 43. Based on calculations from the fusion experiments, the viscosity-to-surface tension ratio was estimated to be approximately 0.044 s/ μm , which is consistent with values reported for other coacervate and condensate systems (Nature Communications, 2020, 11, 4628; PNAS, 2015, 112(23), 7189).

The following discussion has been included in the manuscript:

“The relationship between fusion time (τ) and average droplet radius (l) for two coalescing droplets can be used to approximate of the inverse capillary velocity (η/γ). In this model, the droplets are assumed to be dispersed in a low-viscosity medium, where $\tau \approx l(\eta/\gamma)$. Here, η and γ represent the the viscosity of the coacervate droplets and the interfacial tension, respectively.⁵⁴ For the peptide coacervate droplets, the η/γ value was determined to be $\sim 0.044 \text{ s } \mu\text{m}^{-1}$, which is comparable to those of other peptide-based condensate or coacervate systems (Supplementary Figure 43).⁵⁵ This value is consistent with slow relaxation, indicative of soft and potentially viscoelastic droplets.”

- Also, line 178, the author mention the “formation of relatively stable coacervate droplets”. What is meant by “relatively”?

Response:

Based on our findings, FFM or MFF alone, upon increasing pH, formed coacervate droplets that transformed into irregular aggregates or fibrous structures within 2 minutes of observation. In contrast, the mixture of FFM and MFF formed coacervate droplets that remained stable and did not convert into fibrous structures during prolonged observation. Here, “relatively” refers to the enhanced stability of the coacervate droplets formed by the mixture of FFM and MFF compared to those formed by FFM or MFF alone.

Revised sentenced: *“When the pH was increased to approximately 9, microscopy images revealed the formation of relatively stable coacervate droplets without conversion to fibers or solid aggregates for at least 20 minutes of observation (Supplementary Figure 22).”*

- The FRAP experiments are performed using a low molecular weight dye (rhodamine B). The recovery appears slow for such a small molecule, suggesting low mobility within the droplets. Could the authors please comment on this?

Response:

The recovery of rhodamine B in peptide coacervates in FRAP experiments, despite its low molecular weight, is related to several factors, including droplet viscosity and molecular interactions between the droplets and rhodamine B. For example, the internal microenvironment of the droplets, especially viscosity, could significantly affect the diffusion of small molecules such as rhodamine B. High viscosity, likely due to densely packed peptides, can indeed limit the diffusion of rhodamine B. In addition, rhodamine B, as a typical aromatic fluorophore, is concentrated in the interior of the droplets. High viscosity, probably due to the densely packed peptides, can indeed restrict the diffusion of rhodamine B. In addition, rhodamine B, as a typical aromatic fluorophore, is concentrated inside the droplets with a partition coefficient ($K = F_{\text{droplet}}/F_{\text{background}}$) of ~ 169 . This indicates that rhodamine B has a high possibility of interacting with peptides in the droplet components via aromatic, cation- π , hydrophobic and hydrogen bonding interactions, which could affect its diffusion rate and hinder recovery during FRAP. In addition to the intrinsic properties of the droplets and molecular dyes, technical factors such as dye concentration and bleaching time can also influence the recovery behavior.

ii/ Clarification of the role of electrostatics. The authors mention that “the electrostatic repulsion between the peptides is screened by the electrolyte, favouring phase separation” (lines 241-242), but also explain that phase separation is promoted at pH 9 due to charge neutralization of the amine groups (lines 164 and following). This discussion could benefit from clarification. Providing the pKa of the amine groups in the different peptides used would help resolve this.

Response:

The following sentence has been added to the manuscript:

“The observed behaviour of the short peptides in the solution can be attributed to the charge of the -NH₂ group, which is positive at acidic pH and neutral at basic pH (pka of the α -amino group in F or M amino acid \approx 7).”

iii/ Temperature responsiveness. Could the authors comment on whether the observed behavior depends on the heating rate? Additionally, is there any hysteresis when cooling down the solution?

Response:

The heating rate can significantly influence the kinetics of the coacervate-to-solution transition. Rapid heating may prevent the coacervates from fully equilibrating, leading to abrupt or incomplete transitions, whereas slower heating allows for a more gradual and complete dissolution as the temperature increases, enabling the interactions driving coacervation (e.g., aromatic, electrostatic, hydrophobic, and hydrogen bonding) to weaken progressively. During cooling, hysteresis is likely, as coacervates may not re-form at the same temperature they dissolved due to kinetic barriers or delayed molecular reorganization. Cooling too quickly can result in kinetic trapping, where peptides fail to re-establish interactions necessary for coacervation, leading to smaller, irregular, or incomplete coacervates. Slower, equilibrium-controlled cooling allows sufficient time for peptides to reorganize, enabling more uniform coacervate re-formation.

iv/ Partitioning of dyes, enzymes and enzyme reactions.

- The graph shown in Figure 5g is potentially misleading, as it gives the impression of a trend in the partition coefficients. How are the cargoes classified along the x-axis? I suggest avoiding ranking them by decreasing partition coefficients unless this reflects an intrinsic property of the dyes (e.g., hydrophobicity or charge). Instead, classifying them as neutral, positive, and negative might be clearer.

Response:

In response to the reviewer’s comments, Figure 5g has been revised in the updated manuscript.

- Cy5-GOx appears to accumulate at the droplets interface. Could the authors comment on this observation?

Response:

The accumulation of Cy5-GOx at the peptide coacervate interface is likely influenced by a combination of electrostatic, hydrophobic, and steric factors. The hydrophobic Cy5 fluorophore may interact with the amphiphilic interface of the coacervates, while specific interactions such as hydrogen bonding, hydrophobic effects, or π - π stacking with aromatic residues could also contribute to its localization. Additionally, the dense interior of the coacervates may sterically exclude large macromolecules like GOx, favoring their accumulation at the less dense interface through phase partitioning. This highlights the role of both molecular size and interaction dynamics in its interfacial localization.

- Lines 410-411 state that “Confocal imaging of the reaction revealed the production of fluorescent resorufin ($\lambda_{em}\sim 580$ nm) confined within the coacervates”. However, there is no direct evidence that resorufin is only produced within the droplets. It is possible that the reaction occurs outside, with the product subsequently diffusing and accumulating in the droplets. Could the authors address this point?

Response:

Peptide-based coacervate droplets, as membraneless compartments, concentrate enzymes (e.g. HRP, GOx) and fluorophores (e.g. Nile Red, THT, Rhodamine B) due to their partitioning effects (Figure 5, Supplementary Figures 62/64). The surrounding dilute environment allows a dynamic exchange of components and H₂O₂ is homogeneously dispersed. Thus, enzymatic reactions can take place both inside and outside the coacervates. However, the partitioning effect results in much higher concentrations of enzymes and substrates (e.g. HRP and Amplex Red) within the droplets, increasing reaction efficiency. In addition, coacervates sequester resorufin (partition coefficient 46, Figure 5), resulting in its accumulation in the interior regardless of where it is produced.

To avoid confusion, the sentence in lines 410-411 has been revised to read:

“Confocal imaging of the reaction showed that the fluorescent resorufin product ($\lambda_{em}\sim 580$ nm) was mainly confined within the coacervates (Supplementary Figure 63).”

v/ Comparison with “conventional” coacervates (lines 441-455). The term “conventional” coacervates (lines 441-455) seems ambiguous, and I am not sure whether the coacervates used here as a single example reflect broader trends in complex coacervates. Complex coacervates are known to sequester both hydrophobic dyes and hydrophilic cargoes, and these sequestration properties can be highly specific and system-dependent. For this reason, I would recommend removing this direct comparison between peptide coacervates and the specific amylose coacervates. If necessary, this discussion could be reserved for later when addressing hierarchical coacervate assembly.

Response:

To avoid confusion, the term “conventional complex coacervates” has been replaced:

“The binary peptide coacervates exhibited high resistance to electrolytes, in contrast to polyelectrolytes-based complex coacervates, which typically dissolve in the presence of >500 mM NaCl.⁵⁶”

Additionally, the direct comparison between peptide coacervates and amylose coacervates has been removed. Further discussions have been added in the revised manuscript.